# MASS: MAthematical Data Selection via Skill Graphs for Pretraining Large Language Models

**Jiazheng Li** [* 1] **Lu Yu** [* 2] **Qing Cui** [2] **Zhiqiang Zhang** [2] **Jun Zhou** [† 2] **Yanfang Ye** [3] **Chuxu Zhang** [† 1]

## Abstract

High-quality data plays a critical role in the pretraining and fine-tuning of large language models (LLMs), even determining their performance ceiling to some degree. Consequently, numerous data selection methods have been proposed to identify subsets of data that can effectively and efficiently enhance model performance. However, most of these methods focus on general data selection and tend to overlook the specific nuances of domain-related data. In this paper, we introduce MASS, a **MA**thematical data **S**election framework using the **S**kill graph for pretraining LLMs in the mathematical reasoning domain. By taking into account the unique characteristics of mathematics and reasoning, we construct a skill graph that captures the mathematical skills and their interrelations from a reference dataset. This skill graph guides us in assigning quality scores to the target dataset, enabling us to select the top-ranked subset which is further used to pretrain LLMs. Experimental results demonstrate the efficiency and effectiveness of MASS across different model sizes (1B and 7B) and pretraining datasets (web data and synthetic data). Specifically, in terms of efficiency, models trained on subsets selected by MASS can achieve similar performance to models trained on the original datasets, with a significant reduction in the number of trained tokens - ranging from 50% to 70% fewer tokens. In terms of efficacy, when trained on the same amount of tokens, models trained on the data selected by MASS outperform those trained on the original datasets by 3.3% to 5.9%. These results underscore the potential of MASS to improve both the efficiency and effectiveness of pretraining LLMs.

---

[*]Equal contribution [1]University of Connecticut, USA [2]Ant Group, China [3]University of Notre Dame, USA. Correspondence to: Chuxu Zhang <chuxu.zhang@uconn.edu>, Jun Zhou <jun.zhoujun@antgroup.com>.

*Proceedings of the 42nd International Conference on Machine Learning*, Vancouver, Canada. PMLR 267, 2025. Copyright 2025 by the author(s).

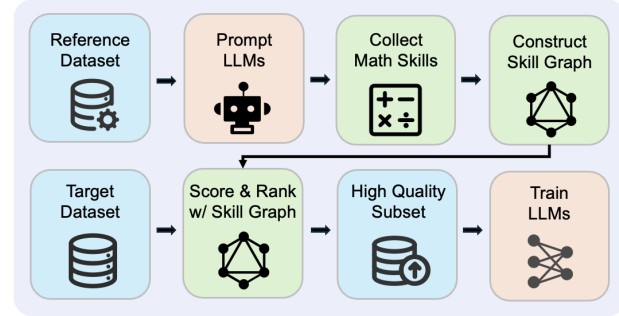

*Figure 1.* The pipeline of MASS. (1) We first employ prompt engineering to extract mathematical skills from a reference dataset, thereby constructing a skill graph. (2) We then score and rank the target dataset using the constructed skill graph. The top-ranked subset can further be selected and used to train LLMs.

## 1. Introduction

The success of large language models (LLMs) is closely tied to the scaling up of model size, training data, and computational resources (Brown, 2020; Hoffmann et al., 2022; Qwen Team, 2024). Among these three factors, training data serves as the foundation of LLMs' rich knowledge and capabilities, enabling them to excel in creative writing, complex reasoning, and even agentic planning. Recently, there is mounting evidence that pretraining on high-quality and diverse corpora can significantly enhance LLM performance (Penedo et al., 2024; Li et al., 2024a). For example, Microsoft's Phi series of models has been renowned for their pretraining corpus's quality and efficiency. The latest Phi-4 model (Abdin et al., 2024), trained on 10T tokens, has surpassed other models of the same size, including Qwen 2.5 (Qwen Team, 2024), which was trained on 18 trillion tokens. This has led many researchers to focus on LLM data curation methodologies, which include the selection of high-quality sub-datasets (Wettig et al., 2024), deduplication of extensive datasets (Tirumala et al., 2023), and the balancing of data from various domains (Xie et al., 2024), among other techniques.

In terms of data selection methods, the objective is to extract high-quality subsets from the original datasets such that training LLMs on these subsets can achieve similar or even superior performance on downstream tasks compared

to training on the entire original datasets. There are two main categories of methods: heuristic-based approaches and model-based approaches. For the former, a variety of manually designed filters, including mean word length, stop word fraction, and word repetitions, are frequently applied to preprocess raw web data (Tirumala et al., 2023). For the latter, models similar to BERT (Devlin et al., 2019) are commonly used to evaluate and select data with higher scores, while other approaches directly employ GPT-like models to assess the data's value.

While the emergence of OpenAI o-series models (OpenAI, 2024), DeepSeek-R1 (DeepSeek-AI et al., 2025), and similar models, focusing on advanced reasoning capabilities, highlights the growing emphasis on the mathematical and reasoning abilities of LLMs, current data selection methods usually concentrate on general domains, neglecting the specific attributes of mathematical domains. AutoDS (Zhang et al., 2024b), as an exception, uses Qwen2-72B to score the OpenWebMath (Paster et al., 2024) dataset based on whether each text demonstrates mathematical intelligence and suitability for educational purposes in mathematics. Although it employs tailored prompts for mathematical texts, it still overlooks the deeper nuances of the mathematical domain when compared to the general domain, particularly from the perspective of math skills.

Therefore, in this paper, we delve into data selection for LLMs and propose MASS - a **MA**thematical data **S**election framework using **S**kill graphs, to enhance their mathematical and reasoning abilities by considering the underlying skills (or knowledge points) and their interrelations embedded in mathematical datasets. ***MASS has two selection guidelines: (1) a data point that encompasses more important math skills should have a higher quality score; and (2) a data point that covers more important compositional information of math skills should have a higher quality score.*** As illustrated in Figure 1, we begin by utilizing a high-quality reference dataset and instructing a strong LLM to extract embedded math skills. These skills are then used to construct a skill graph, where nodes represent skills and edges denote the co-occurrence of skills. Next, we calculate the semantic similarities between these math skills and the target dataset to be selected. We further aggregate these similarities on the graph structure to integrate the compositional information of the skills to obtain final quality scores for each data point in the target dataset. Finally, these scores are ranked, allowing us to select a high-quality subset from the original dataset.

In this work, we select NuminaMath (LI et al., 2024) as the reference dataset to construct a skill graph based on its mathematical skills. We then apply our method to three target datasets: two web datasets (OpenWebMath (Paster et al., 2024), OpenWebMath-pro (Zhou et al., 2024a)) and one

synthetic dataset (Jiuzhang3.0-CoT (Zhou et al., 2024c)). Using the skill graph, we identify and extract high-quality subsets from each of these target datasets. We continue pretraining `TinyLlama-1.1B` and `Mistral-7B` on the original and selected subsets. Experimental results demonstrate that with merely 50% to 70% training steps, models trained on selected subsets can achieve similar performances than those using original datasets. When trained on the same amount of tokens, models trained on selected subsets can outperform those using original datasets by 3.3% to 5.9%. This highlights the efficiency and efficacy of our data selection method in enhancing the mathematical and reasoning capabilities of LLMs.

## 2. The Proposed MASS

### 2.1. Overview

Given a target dataset $\mathcal{D}_{tgt} = \{x_1, x_2, \cdots, x_N\}$ where each data point $x$ is a piece of text, our goal is to select a high quality subset $\mathcal{D}_{hq}$ with budget $b \in \mathbb{Z}^+$, where $b = |\mathcal{D}_{hq}| \ll |\mathcal{D}_{tgt}|$. When continuing pre-training or fine-tuning LLMs on $\mathcal{D}_{hq}$, we can achieve similar or even superior downstream performances compared to training on $\mathcal{D}_{tgt}$. As illustrated in Figure 2, our proposed MASS can be divided into two main steps. In the first step, we collect a small-scale high quality reference dataset $\mathcal{D}_{ref}$ and instruct a strong LLM to identify the mathematical skills present in $\mathcal{D}_{ref}$. Next, we construct a skill graph $G_{skill}$ that encapsulate the compositional relationships among the various skills. In the second step, we then use $G_{skill}$ to guide data selection. Specifically, we compute the similarities between the target dataset and the extracted skills. We then perform aggregation and pooling through the skill graph to generate final quality scores for each data point in the target dataset. These scores are eventually used to rank and select the high-quality subset $\mathcal{D}_{hq}$. Details of these two steps are described in the following.

### 2.2. Skill Graph Construction

**Skills Extraction.** We randomly sample 100K data points from NuminaMath dataset (LI et al., 2024) as the reference data $\mathcal{D}_{ref} = \{d_1, d_2, \cdots, d_R\}$ where $|\mathcal{D}_{ref}| = 100K$. NuminaMath, which comprises 860K pairs of competition math problems and their solutions, is considered to be of high quality (i.e., each text is meticulously collected and annotated with accompanying chain-of-thought traces) and diverse (i.e., it includes problems ranging from high-school-level to advanced-competition-level). We then instruct `Qwen2.5-72B-Instruct-GPTQ-Int4` (Qwen Team, 2024) to identify 1 to 10 tested mathematical skills in each data point of the reference dataset due to its excellent accuracy and latency. The prompt template we use is shown in the Appendix C. As a result, we collect 865K

skills from $\mathcal{D}_{ref}$ in total. An example of the data sample and its corresponding skills is shown below.

---

**Skill Extraction Example**

**CONTENT:** Find the greatest value of $x$ such that $\frac{x^2-x-90}{x-9} = \frac{2}{x+7}$. The expression when simplified by factoring the numerator transforms into: $\frac{(x-9)(x+10)}{x-9} = \frac{2}{x+7}$. Canceling the $(x-9)$ factor on both sides, provided $x \neq 9$, we get: $x + 10 = \frac{2}{x+7}$. Multiplying both sides by $(x + 7)$ to eliminate the fraction yields: $(x + 10)(x + 7) = 2$. Expanding and rearranging this equation results in: $x^2 + 17x + 70 = 2 \implies x^2 + 17x + 68 = 0$. Factoring the quadratic gives: $(x+4)(x+13) = 0$. The solutions to this equation are $x = -4$ and $x = -13$. The greatest of these solutions is $\boxed{-4}$.

- - - - - - - - - - - - - - - - - - - - - - - - - - - - - -

**SKILLS:** Equation solving, Factoring polynomials, Fraction manipulation, Quadratic equations, Root identification, Expression simplification, Algebraic transformation, Polynomial division, Inequality consideration, Solution verification

---

**Skill Graph Construction.** Once we have identified a sufficient set of mathematical skills, we construct a skill graph $G_{skill} = (V, E, \mathbf{A})$ where $V$ represents node set of skills, $E$ denotes the edge of skill co-occurrence, and $\mathbf{A}$ is the adjacency matrix. To reduce noise, we preprocess these extracted skills by merging those with a semantic similarity greater than 0.9, utilizing `bge-large-en-v1.5` (Xiao et al., 2024) as the text encoder. As a result, 46,490 unique nodes (skills) are retained to form the node set $V = \{v_1, v_2, \cdots, v_{|V|}\}$ ($|V| = 46,490$). Each node $v_i$ has two attributes: (1) $v_i^{cnt}$, a scalar indicating the skill's number of occurrence; (2) $v_i^{ids}$, a list containing the indices of the sub-reference dataset where this skill appears. For edges, we create an edge between two skills when they co-occur in at least one data point, resulting in 1,184,007 edges in total, forming the edge set $E = \{e_{ij} = (v_i, v_j) \mid f_{co}(v_i, v_j) > 0\}$, where $f_{co}(v_i, v_j)$ denotes the co-occurrence count of $(v_i, v_j)$. Each edge $e_{ij}$ has an attribute: $e_{ij}^{cnt}$, a scalar indicating the number of co-occurrences of the skill pair. Next, we define the adjacency matrix $\mathbf{A} \in \mathbb{R}^{|V| \times |V|}$ for the skill graph. Specifically, for diagonal elements $\{\mathbf{A}_{i,i} \mid i = 1, 2, \cdots, |V|\}$, the values are normalized from the number of occurrence of the skill $v_i$ using softmax function with temperature coefficient $T$:

$$\mathbf{A}_{i,i} = \sigma(v_i^{cnt}, T) = \frac{\exp\left(\frac{v_i^{cnt}}{T}\right)}{\sum_{j=1}^{|V|} \exp\left(\frac{v_j^{cnt}}{T}\right)}. \quad (1)$$

For non-diagonal elements $\{\mathbf{A}_{i,j} \mid i \neq j, i, j =$

$1, 2, \cdots, |V|\}$, the values are similarly normalized from the number of occurrence of the skill pair $(v_i, v_j)$:

$$\mathbf{A}_{i,j} = \sigma(e_{ij}^{cnt}, T) = \frac{\exp\left(\frac{e_{ij}^{cnt}}{T}\right)}{\sum_{(e_{ij} \in E)} \exp\left(\frac{e_{ij}^{cnt}}{T}\right)}. \quad (2)$$

By far, we have constructed the skill graph $G_{skill}$ that encapsulates the skills tested in the reference dataset and their compositional relationships. A visualization of a sub-skill graph and more statistics of the entire graph can be found in Appendix B.

### 2.3. Data Selection via Skill Graph

With the reference dataset $\mathcal{D}_{ref} = \{d_1, d_2, \cdots, d_R\}$ and the skill graph $G_{skill}$ constructed in the previous step, we propose to select a high quality subset $\mathcal{D}_{hq}$ from the target dataset $\mathcal{D}_{tgt} = \{x_1, x_2, \cdots, x_N\}$. The detailed process is outlined as follows.

**Semantic Similarities Computation.** Given a target data point from target dataset $x_i \in \mathcal{D}_{tgt}$, and a specific skill from the skill set $v_j \in V$, we compute their semantic similarity:

$$\text{sim}(x_i, v_j) = \max_{k \in v_j^{ids}} \cos(\text{Emb}(x_i), \text{Emb}(d_k)), \quad (3)$$

where $\cos$ denotes the cosine similarity, $v_j^{ids}$ is the list of indices of the sub-reference dataset where this specific skill $v_j$ appears, $d_k$ is the $k$-th data point in the reference dataset and $\text{Emb}()$ is the text embedding model (we use `bge-large-en-v1.5` (Xiao et al., 2024) in this work). The intuition behind this approach is to calculate the cosine similarity between the target data $x_i$ and each of the corresponding reference dataset $\{d_k \mid k \in v_j^{ids}\}$ of the node (skill) $v_j$. The maximum similarity value is then selected as the similarity score $\text{sim}(x_i, v_j)$. Similarly, we compute the similarities between each data point of the target dataset and each of the skills, forming the similarity matrix $\mathbf{S} \in \mathbb{R}^{N \times |V|}$, where $N$ is the size of the target dataset and $|V|$ is the number of nodes in the skill graph.

**Aggregation on Skill Graph.** With the above formulation of $\text{sim}(x_i, v_j)$, we aggregate the computed similarities on the skill graph using the adjacency matrix $\mathbf{A}$. The aggregated similarity $\text{sim}_{agg}(x_i, v_j)$ between the target data $x_i$ and the skill $v_j$ is calculated as:

$$\begin{aligned}
\text{sim}_{agg}(x_i, v_j) = {} & \mathbf{A}_{j,j}\text{sim}(x_i, v_j) \\
& + \sum_{v_k \in \mathcal{N}(v_j)} \mathbf{A}_{j,k}\text{sim}(x_i, v_k), \quad (4)
\end{aligned}$$

where $\mathcal{N}(v_j)$ denotes the neighbor set of $v_j$ in the skill graph. The intuition is that the aggregated similarity $\text{sim}_{agg}(x_i, v_j)$ is not solely based on the original similarity $\text{sim}(x_i, v_j)$ and the skill importance $\mathbf{A}_{j,j}$, but also in-

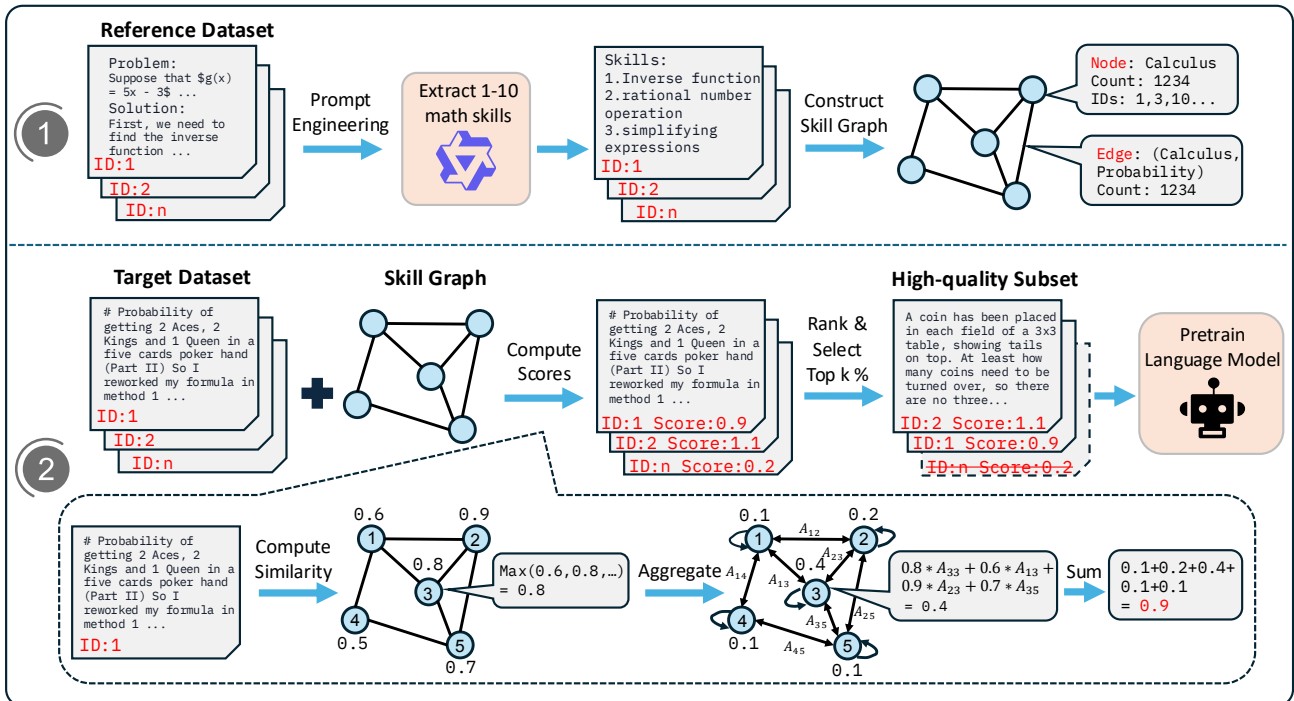

*Figure 2.* **An illustration of MASS.** In the first step, we instruct `Qwen2.5-72B-Instruct-GPTQ-Int4` (Qwen Team, 2024) to extract 1 to 10 mathematical skills from each data point in the reference datasets. Using these extracted skills, we construct a skill graph that captures the compositional relationships among them. In the second step, we assign a quality score to each data point in the target dataset. This score is calculated by pooling and aggregating the similarities of the data point and the skills in the skill graph. Finally, the top-ranked data points can be selected for training the large language models.

cludes the weighted sum of the similarities of the connected skills, incorporating the compositional relationships of these skills. In practice, we can compute the aggregated similarity $\text{sim}_{agg}(x_i, v_j)$ efficiently using matrix multiplication:

$$\mathbf{S_{agg}} = \mathbf{SA}, \quad (5)$$

where $\mathbf{S_{agg}}$ and $\mathbf{S}$ represent the aggregated and the vanilla similarity matrices respectively, with $\mathbf{A}$ being the adjacency matrix. For each target data point $x_i$, its quality score is obtained by summing the similarities between $x_i$ and all the skills $v_j$, which is formulated as:

$$\text{score}(x_i) = \sum_{j=1}^{|V|} \text{sim}_{agg}(x_i, v_j). \quad (6)$$

Next, we can compute the score vector $\mathbf{score}(\mathcal{D}_{tgt})$ for the entire target dataset $\mathcal{D}_{tgt} = \{x_1, x_2, \cdots, x_N\}$ in matrix form as follows:

$$\begin{aligned} \mathbf{score}(\mathcal{D}_{tgt}) &= [\text{score}(x_1), \text{score}(x_2), \cdots, \text{score}(x_N)] \\ &= \sum_{j=1}^{|V|} \mathbf{S_{agg}}_{:,j}. \end{aligned}$$
$$(7)$$

**Data Selection.** Given the score vector $\mathbf{score}(\mathcal{D}_{tgt})$ and a selection ratio $k\%$, we determine whether each target data point $x_i$ should be retained or discarded as follows:

$$\text{op}(x_i, k) = \begin{cases} \text{keep}(x_i) & \text{if } \text{score}(x_i) \text{ ranks in the top } k\% \\ & \text{of } \mathbf{score}(\mathcal{D}_{tgt}), \\ \text{drop}(x_i) & \text{otherwise.} \end{cases}$$
$$(8)$$

Finally, by applying $\text{op}(x_i, k)$ to to all data points in the target dataset $\mathcal{D}_{tgt}$, we obtain the high quality subset $\mathcal{D}_{hq}$:

$$\mathcal{D}_{hq} = \{\text{op}(x_1, k), \text{op}(x_2, k), \cdots, \text{op}(x_N, k)\}. \quad (9)$$

### 2.4. Why MASS Works?

As stated in the introduction, the goal of our method is to ensure that each data point receives a higher score if it satisfies two key conditions: (1) it covers a broader range of important mathematical skills, and (2) it captures a richer, more nuanced compositional understanding of these skills. These two principles serve as the foundation for evaluating and ranking the quality of data points within the target dataset. Here we provide an explanation of how MASS operates based on these two foundational principles.

Given the adjacency matrix of the skill graph $\mathbf{A}$, according

to Eq. 4 and Eq. 6, the quality score of a given data point $x$ consists of two components:

$$\text{score}(x) = \sum_{j=1}^{|V|} \text{sim}_{agg}(x, v_j)$$

$$= \sum_{j=1}^{|V|} \mathbf{A}_{j,j}\text{sim}(x, v_j) + \sum_{j=1}^{|V|} \sum_{v_k \in \mathcal{N}(v_j)} \mathbf{A}_{j,k}\text{sim}(x, v_k). \quad (10)$$

The first component is a weighted sum of similarities to these skills, where the weights are given by the diagonal elements of the adjacency matrix. As described in Eq. 1, if a skill $v_j$ is more important (i.e., it appears more frequently), the corresponding element $\mathbf{A}_{j,j}$ has a larger value. Consequently, the more important skills a data point covers, the greater the value of the first component.

Similarly, the second component is also a weighted sum of similarities, where the weights correspond to the non-diagonal elements of the adjacency matrix, which encode the compositional relationships between skills. According to Eq. 2, if a skill pair $(j, k)$ is more important (it appears more frequently), the corresponding element $\mathbf{A}_{j,k}$ has a larger value. Therefore, the more important compositional information a data point covers, the larger the second component becomes.

## 3. Experiments

### 3.1. Experiment Setup

**Training Corpora.** We utilize different types of pretraining datasets, including math-related web data OpenWeb-Math (Paster et al., 2024), OpenWebMath-pro (Zhou et al., 2024a) and synthetic data Jiuzhang3.0 (Zhou et al., 2024c). OpenWebMath is a dataset that encompasses a large portion of mathematical text from the internet. It has been filtered and extracted from over 200 billion HTML files on Common Crawl, resulting in a refined set of 6.3 million documents containing a total of 14.7 billion tokens. OpenWebMath-pro is an enhanced version of OpenWebMath, further refined using the ProX framework, and contains approximately 5 billion high-quality tokens. Jiuzhang3.0-Corpus-PT-CoT is a synthetic dataset composed of 3.8 billion high-quality math-related tokens, including question-answer pairs sourced from a variety of origins.

**Training Setup.** We continue pretraining (CPT) `TinyLlama-1.1B` (Zhang et al., 2024a) and `Mistral-7B` (Jiang et al., 2023) using both original datasets and the selected subsets identified by various methods. To accelerate the training process, we leverage DeepSpeed Zero-2 (Ren et al., 2021) and Flash Attention 2 (Dao, 2023). The specific hyperparameter settings used in

*Table 1.* Hyperparameter settings.

| Hyperparameter | `TinyLlama-1B` | `Mistral-7B` |
|---|---|---|
| Peak Learning Rate | 8e-5 | 2e-5 |
| Context Length | 2,048 | 4,096 |
| Batch Size | 512 | 256 |
| Learning Rate scheduler | Cosine Annealling | |
| Optimizer | AdamW | |
| Warmup Ratio | 0.01 | |
| Weight Decay | 0.1 | |

our experiments are provided in Table 1.

**Baselines.** We compare MASS with various data selection baselines which are from different method categories. They include (1) AutoDS (Zhang et al., 2024b): This method utilizes meta-prompted language models as zero-shot verifiers to autonomously evaluate and select high-quality mathematical content. (2) DSIR (Xie et al., 2023): This approach estimates importance weights in a reduced feature space for tractability and selects data through importance resampling based on these weights. (3) RHO-1 (Lin et al., 2024): This method selects suitable tokens during the training process using selective language modeling and training dynamics. (4) ProX (Zhou et al., 2024a): This approach refines pretraining data at scale through program generation using language models. (5) BM25 (Robertson & Zaragoza, 2009): This method ranks and selects documents based on query term occurrence and rarity across the corpus that extends TF-IDF by considering term frequency saturation and document length. (6) Heuristic rules used in the construction of FineWeb corpora (Penedo et al., 2024): This method employs heuristic rules for data selection. Note that we mainly implement these baselines for `TinyLlama-1.1B` considering the compute resources.

**Evaluation Setup.** We compare the model performances on nine mathematics-related benchmarks used in RHO-1 (Lin et al., 2024) and ProX (Zhou et al., 2024a). The evaluation is conducted using the same implementation and includes few-shot chain-of-thought (CoT) examples.

The code is available: github.com/lijiazheng0917/MASS.

### 3.2. Main Results

The main performance comparison results are presented in Table 2. On average, across nine math and reasoning downstream tasks, the vanilla `TinyLlama-1.1B` and `Mistral-7B` achieve performance scores of 13.3% and 53.0%, respectively. To assess the effectiveness of our approach, we continue pretraining these two models using both the original datasets and the selected subsets. Specifically, for OpenWebMath, OpenWebMath-pro, and Jiuzhang3.0, we apply MASS to select the top 30%, top 60%, and top 70% of tokens, respectively, based on the varying qualities

*Table 2.* The main experimental results. `TinyLlama-1.1B` and `Mistral-7B` are continuously pretrained using both the original and selected subsets of OpenWebMath, OpenWebMath-pro, and Jiuzhang3.0. The **bolded** entries indicate the best results within each setting. * indicates that results are from ProX (Zhou et al., 2024a)

| Dataset | Method | Unique Tokens | Trained Tokens | GSM8K | MATH | SVAMP | ASDiv | MAWPS | TAB | MQA | MMLU STEM | SAT MATH | Avg. |
|---|---|---|---|---|---|---|---|---|---|---|---|---|---|
| | | | | | | `TinyLlama-1.1B` | | | | | | | |
| w/o continual pretraining | | | | 2.7 | 2.8 | 10.9 | 17.9 | 20.5 | 12.5 | 14.0 | 16.3 | 21.9 | 13.3 |
| OpenWebMath | - | 14.6B | 14.6B | 5.2 | 3.0 | 20.7 | 31.4 | 41.0 | 14.6 | 10.1 | 19.5 | **37.5** | 20.3 |
| | RULE* | 6.5B | 15B | 4.5 | 2.8 | 17.5 | 29.4 | 39.3 | 15.1 | 12.4 | 19.4 | 25.0 | 18.4 |
| | RHO-1* | 14.6B | 9B | 7.1 | **5.0** | 23.5 | 41.2 | 53.8 | - | **18.0** | - | - | - |
| | ProX | 5.1B | 14.6B | 8.6 | 3.0 | 23.8 | 40.2 | 51.6 | 19.6 | 14.9 | **26.1** | 25.0 | 23.6 |
| | DSIR | 4.9B | 14.6B | 5.5 | 2.6 | 24.1 | 37.8 | 54.3 | 16.9 | 12.1 | 25.4 | 22.3 | 22.1 |
| | AutoDS | 4.9B | 14.6B | 7.3 | 2.4 | 22.9 | 39.2 | 52.7 | 18.4 | 13.8 | 23.2 | 24.1 | 22.7 |
| | MASS | 4.9B | 14.6B | **9.0** | 4.4 | **24.9** | **41.4** | **54.8** | **21.5** | 13.9 | 20.3 | 25.0 | **23.9** |
| OpenWebMath -pro | - | 5.1B | 14.6B | 8.6 | 3.0 | 23.8 | 40.2 | 51.6 | 19.6 | 14.9 | 26.1 | 25.0 | 23.6 |
| | DSIR | 3B | 14.6B | 8.8 | 3.2 | **24.1** | 41.5 | 53.1 | 18.9 | 14.3 | **27.6** | 27.5 | 24.4 |
| | AutoDS | 3B | 14.6B | 9.1 | 4.5 | 22.4 | 40.8 | 54.3 | 23.2 | 13.1 | 26.5 | 28.0 | 24.7 |
| | MASS | 3B | 14.6B | **10.2** | **5.8** | 23.8 | **42.3** | **57.9** | **25.3** | **15.3** | 27.0 | **34.4** | **26.9** |
| Jiuzhang3.0 | - | 3.4B | 6.8B | 22.3 | 19.0 | 46.4 | 60.1 | 73.2 | 29.6 | 19.1 | **24.0** | 34.4 | 36.4 |
| | DSIR | 2.4B | 6.8B | 24.5 | 21.3 | 48.2 | 63.9 | 74.4 | 28.8 | 19.2 | 22.1 | 33.6 | 37.3 |
| | AutoDS | 2.4B | 6.8B | 26.7 | 20.8 | 51.3 | 66.7 | 73.5 | 31.1 | 19.3 | 22.4 | 32.8 | 38.3 |
| | MASS | 2.4B | 6.8B | **30.1** | **24.8** | **52.5** | **69.1** | **80.7** | **32.9** | **20.4** | 22.7 | **34.4** | **40.8** |
| | | | | | | `Mistral-7B` | | | | | | | |
| w/o continual pretraining | | | | 41.1 | 10.6 | 64.9 | 68.5 | 87.3 | 54.8 | 33.9 | 49.9 | 65.6 | 53.0 |
| OpenWebMath | - | 14.4B | 9.6B | 44.5 | 19.0 | 60.6 | 68.4 | 87.8 | 50.5 | 44.5 | 50.9 | 56.2 | 53.6 |
| | MASS | 4.8B | 9.6B | **47.7** | **23.2** | **64.6** | **74.7** | **90.5** | **55.7** | **50.7** | **52.6** | **65.6** | **58.4** |
| OpenWebMath -pro | - | 5.1B | 5.1B | 47.1 | 21.8 | 63.2 | 73.7 | 89.5 | **58.2** | 42.6 | 52.2 | 56.2 | 56.1 |
| | BM25 | 3B | 5.1B | 44.7 | 24 | 63.1 | 73 | 86.1 | 49.1 | 49.8 | 52.6 | 67.1 | 56.6 |
| | DSIR | 3B | 5.1B | 42.1 | 21.6 | 63.6 | 73.4 | 86.8 | 50.4 | **55.3** | 51.9 | 70.8 | 57.5 |
| | MASS | 3B | 5.1B | **53.2** | **25.6** | **67.0** | **76.8** | **90.4** | 57.6 | 51.8 | **54.5** | **81.2** | **62.0** |
| Jiuzhang3.0 | - | 3.8B | 3.8B | 66.4 | 39.4 | 82.9 | **85.9** | 90.8 | 35.3 | 61.8 | 40.1 | 50.0 | 61.4 |
| | MASS | 2.7B | 3.8B | **70.0** | **43.8** | **84.3** | 85.7 | **93.7** | **35.7** | **63.5** | **46.9** | **65.6** | **65.5** |

of the datasets. The selected data are repeated proportionally to ensure that the total number of tokens used for training remains consistent.

Table 2 demonstrates that all of these math datasets enhance the models' mathematical reasoning capabilities while our approach achieves the best performance across all these methods. Moreover, compared to the original datasets, using MASS-selected datasets further improves the results by 3.6%, 3.3%, and 4.2% for `TinyLlama-1.1B`, and 4.8%, 5.9%, and 4.1% for `Mistral-7B` on OpenWebMath, OpenWebMath-pro, and Jiuzhang3.0, respectively. Additionally, we present the performance of the intermediate checkpoints trained on both the original and selected datasets, as shown in Figure 3. Clearly, in terms of efficiency, our selected datasets achieve the same results as the original dataset, but with 43%-71% fewer trained tokens. In terms of effectiveness, our approach outperforms the baseline by 3.2%-5.9%.

What distinguishes MASS from other baselines is its fine-grained, skill-centric selection approach. While conventional methods focus on superficial linguistic quality (e.g.,

grammatical correctness, noise reduction, or textual coherence), MASS operates at a deeper semantic level. By analyzing reference data, MASS identifies the specific skills a model requires or lacks and strategically curates training subsets to address these gaps. This targeted skill-based selection contrasts sharply with existing approaches, which often overlook semantic efficacy in favor of surface-level text quality.

### 3.3. Analysis of MASS

In this section, we delve into an in-depth analysis of our data selection approach.

#### 3.3.1. COMPUTE EFFICIENCY OF MASS

We conduct a detailed breakdown of the computational costs associated with MASS, using `OpenWebMath` as the target dataset and `Mistral-7B` as the target model. As summarized in Table 3, the pre-processing steps (including skill extraction, embedding, graph construction, and subset selection) account for less than 3% of the total computational budget, with the majority allocated to the actual model train-

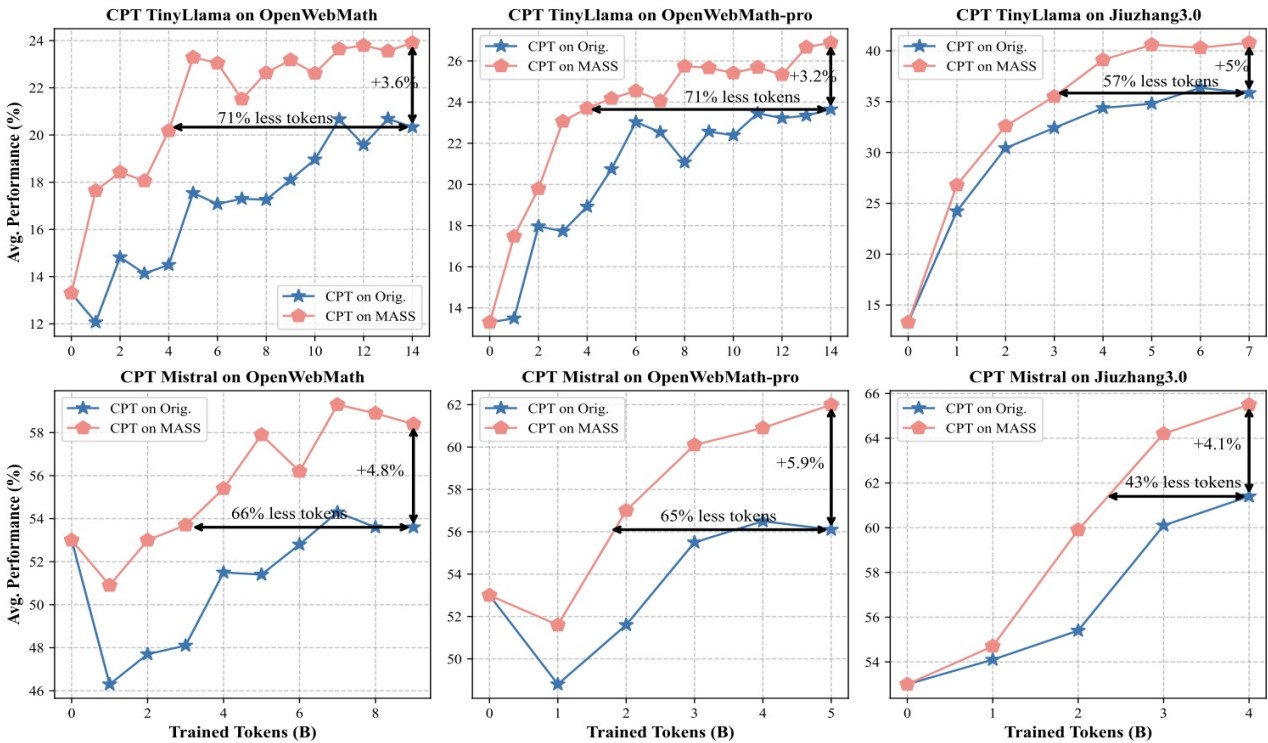

*Figure 3.* The average performance of continual pretrained `TinyLlama-1.1B` and `Mistral-7B` with respect to the number of trained tokens on OpenWebMath, OpenWebMath-pro and Jiuzhang3.0 datasets.

ing phase. This lightweight pre-processing overhead makes MASS highly practical for large-scale applications.

In comparison, existing baselines either do not disclose their pre-processing costs or rely solely on CPU-based operations, making it difficult to perform a precise, end-to-end computational comparison. However, in Figure 3 of our paper, it is clear that MASS achieves more than 40% greater efficiency than all baselines, which means even if we do not consider the pre-processing steps of MASS, it remains the most efficient and effective approach.

### 3.3.2. QUALITY OF DATASETS

Since we have computed the quality score for each data point in the target datasets, we randomly sample 10K data points from each and present the score distributions in Figure 4. It is clear that Jiuzhang3.0 is the highest-quality dataset, with an average score of 1.07, followed by OpenWebMath-pro at 0.95, and OpenWebMath at 0.92. This ranking aligns with the model performance results shown in Table 2. Despite being trained on only 6.8 billion tokens, Jiuzhang3.0 achieves an average performance of 36.4%, significantly outperforming OpenWebMath (20.3%) and OpenWebMath-pro (23.6%), which were trained on approximately 15 billion tokens. This highlights the effectiveness of synthetic data in the pretraining of large language models, demonstrating that even with a smaller token count, synthetic data can drive

superior performance. We provide case studies of the data points with the highest and lowest scores from these three datasets in the Appendix D.

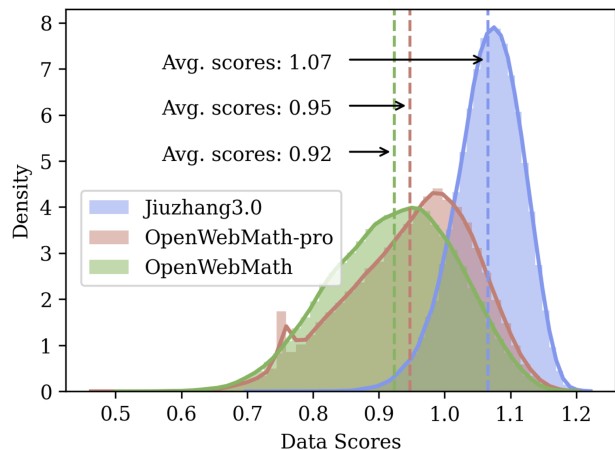

*Figure 4.* Distribution of scores of different datasets.

### 3.3.3. IMPACT OF SELECTION RATIO

Given the varying quality of the datasets, we conduct a series of experiments to examine the impact of the data selection ratio in our method. We continue pretraining `TinyLlama-1.1B` using subsets of OpenWebMath-pro and Jiuzhang3.0 selected at different ratios. The subsets are

*Table 3.* Compute efficiency of MASS

| Operations | A100 GPU Hours |
| --- | --- |
| Extracting skills from reference dataset | $\sim$24 |
| Embedding reference dataset and target dataset | $\sim$0.5 |
| Constructing the skill graph | $\sim$2 CPU hours |
| Selecting a high-quality subset from the target dataset | $\sim$4 |
| Training `Mistral-7B` on the high-quality subset ($\sim$10B tokens) | $\sim$960 |

repeated to reach a total of 15.3 billion and 6.8 billion tokens, respectively, as reported in Table 2. The results, presented in Figure 5, clearly demonstrate that for Jiuzhang3.0, a relatively higher selection ratio leads to higher performance, while for OpenWebMath-pro, a relatively lower selection ratio leads to higher performance. This pattern aligns with the quality of the datasets, indicating that for higher-quality datasets, maintaining a higher selection ratio is more advantageous. Performance trends exhibit a spike-shaped pattern, which can be explained by the quality-diversity tradeoff: (1) at very low selection ratios, the high quality of the selected data comes at the cost of reduced diversity, which negatively impacts model performance. Prior research (Zhang et al., 2025) has consistently highlighted the critical role of diversity in data selection methods; and (2) at excessively high selection ratios, the inclusion of excessive noisy and low-quality data degrades performance.

### 3.3.4. IMPACT OF SKILL GRAPH

As discussed in Section 2.4, a key step in our method for adhering to the two principles is the aggregation of computed similarities on the skill graph. This aggregation enables the incorporation of both the importance of individual skills and their compositional relationships. To assess the effectiveness and necessity of this step, we perform an ablation study with two variants.

*(1) w/o Diag.* In the adjacency matrix $\mathbf{A}$, we set the diagonal elements $\{\mathbf{A}_{i,i} \mid i = 1, 2, \cdots, |V|\}$ to 0. This means that we do not account for our first principle, which pertains to the importance of individual skills.

*(2) w/o Non-diag.* In the adjacency matrix $\mathbf{A}$, we set the non-diagonal elements $\{\mathbf{A}_{i,j} \mid i \neq j, \ i, j = 1, 2, \cdots, |V|\}$ to 0. This indicates that we do not take into account our second principle, which emphasizes the importance of the compositional relationships among skills.

From the results in Table 4, we observe that performance consistently declines when any part of the adjacency matrix is omitted, highlighting that both of our principles play a crucial role in the data scoring and selection process. Additionally, non-diagonal elements have a more significant impact, as indicated by a performance decrease of 2.8% and 2.1% with their omission, compared to 0.4% and 1.2% for

diagonal elements. This could be because more important nodes tend to have more connections with other nodes in the graph. As a result, even in the absence of the diagonal elements, the compositional information derived from the non-diagonal elements can still provide valuable insights into skill importance. This allows the model to retain some level of understanding of the skills' significance, which helps mitigate the performance drop.

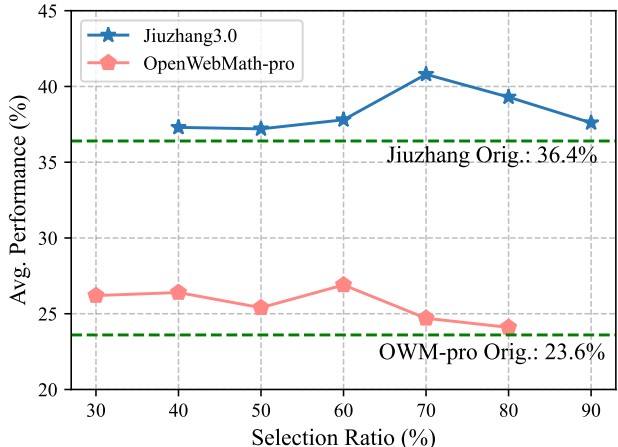

*Figure 5.* The average performance of `TinyLlama-1.1B` with respect to different selection ratios. Note that models are trained on the same amount of tokens.

*Table 4.* The average performance of `TinyLlama-1.1B` with respect to different adjacency matrix forms.

| Method | OpenWebMath-pro | Jiuzhang3.0 |
| --- | --- | --- |
| MASS | 26.9 | 40.8 |
| w/o Diag. | 26.5(0.4%↓) | 39.6(1.2%↓) |
| w/o Non-diag. | 24.1(2.8%↓) | 37.2(2.1%↓) |

### 3.3.5. IMPACT OF SIMILARITY CALCULATION

In Eq. 3, we define the similarity between a data point and a skill by taking the maximum value of the cosine similarities. Below we present two alternative approaches:

*(1) Mean Embedding.* Compute the mean embedding for the relevant sub-reference dataset associated with a skill, and define the similarity as the cosine similarity between

the skill's mean embedding and the data point embedding:

$$\text{Emb}(n_j) = \underset{k \in v_j^{ids}}{\text{mean}} \text{Emb}(d_k), \tag{11}$$

$$\text{sim}(x_i, v_j) = \cos(\text{Emb}(x_i), \text{Emb}(v_j)). \tag{12}$$

*(2) Name Embedding.* Use the embedding of the skill name directly, rather than the corresponding sub-reference dataset, and define the similarity as the cosine similarity between the skill name embedding and the data point embedding:

$$\text{sim}(x_i, v_j) = \cos(\text{Emb}(x_i), \text{Emb}(\text{name}(v_j))). \tag{13}$$

From the results in Table 5, we observe that both alternatives reduce model performance, but to different extents across the two datasets. For OpenWebMath-pro, the average decrease is 0.6%, while for Jiuzhang3.0, the average decline is 3.2%. This discrepancy might be attributed to the dataset format. Specifically, Jiuzhang3.0 is a question-answer pair format, similar to the reference dataset NuminaMath, whereas OpenWebMath-pro consists of web-based math-related data. This may result in less discriminative similarities between a data point and the sub-reference dataset of a skill. As a result, the use of mean embeddings or name embeddings has has a greater impact on the Jiuzhang3.0 dataset compared to OpenWebMath-pro.

*Table 5.* The average performance of `TinyLlama-1.1B` w.r.t. different similarity calculation methods.

| Method | OpenWebMath-pro | Jiuzhang3.0 |
|---|---|---|
| MASS | 26.9 | 40.8 |
| Mean Embedding | 26.4(0.5%↓) | 37.4(3.4%↓) |
| Name Embedding | 26.2(0.7%↓) | 37.8(3.0%↓) |

## 4. Related Work

**See Appendix A for detailed related work discussion.**

Data selection for LLM training falls into heuristic-based and model-based approaches. Heuristic methods, such as language filtering, and deduplication, serve as preprocessing steps, while model-based methods further refine data by improving quality, diversity (Li et al., 2024b; Liu et al., 2024). However, despite the variety of methods available, they often focus on the general domain and neglect the specific aspects of mathematical reasoning. Recent work (Zhang et al., 2024b) evaluates text quality for mathematical intelligence using LLMs, whereas our study approaches the problem from a skill-based perspective.

Mathematical reasoning is crucial for assessing LLMs' intelligence (Shao et al., 2024). High-quality datasets play a key role, such as OpenMathInstruct-2 (Toshniwal et al., 2025) and MathPile (Wang et al., 2024), which offer curated math text. Furthermore, our approach leverages a mathematical skill graph to efficiently select higher-quality data.

Graph-based methods are increasingly valuable in LLM data curation. Skill-it (Chen et al., 2024) demonstrates that LLMs learn skills in a natural order and constructs a skills graph for efficient training. MathScale (Tang et al., 2024) builds a concept graph and uses random walk for instruction data generation. Unlike these approaches, our work focuses on using skill graphs for data selection during continual pretraining to enhance LLMs' reasoning abilities.

## 5. Implications and Future Works

A key distinction between LLMs' mathematical and reasoning capabilities, compared to domains like academic writing or role-playing, is that they can be analyzed through the lens of specific underlying skills. The success of MASS highlights the advantages of incorporating these skills and their compositional relationships into the data selection process, rather than relying on general selection methods.

There are several potential directions for improving our method in the future. For domains with distinct and well-defined skills, such as coding or biomedicine, MASS could be naturally suitable and helpful. However, adapting MASS for areas like creative writing or role-playing, where skills are less clear-cut, remains a significant challenge. Also, to enhance the quality and correctness of extracted skills, we could filter generated skills using a predefined taxonomy. Another crucial aspect not yet addressed by MASS is diversity. While this paper focuses on a continual pretraining paradigm, future applications in small-scale data selection for instruction tuning or reinforcement learning will necessitate combining skills with greater diversity. Additionally, incorporating skill difficulty could boost the model's performance. Currently, relying solely on skill frequency as an importance metric might be limiting; less frequent but more complex skills may require more data for large language models to effectively master.

## 6. Conclusion

In this paper, we propose MASS, a novel and data-efficient framework designed to enhance the mathematical capabilities of large language models (LLMs) by leveraging skill graph-based data selection. First, we employ prompt engineering techniques to extract and categorize mathematical skills from a high-quality reference dataset, constructing a structured skill graph. This graph is then used to score and rank the target dataset, selecting the top-ranked subset for training LLMs. Our empirical results show that MASS significantly improves both the efficiency and effectiveness of the training process, leading to models with stronger mathematical reasoning abilities and faster convergence. This approach offers a more data-efficient way to enhance LLMs, particularly in specialized domains like mathematics.

## Impact Statement

This paper presents work whose goal is to advance the data efficiency of large language model pretraining, which could facilitate wider accessibility of AI technologies and contribute to more sustainable AI development practices. There are no negative societal consequences of our work, as it is specifically designed to improve the data efficiency of existing technologies without introducing any harmful ethical, privacy, or social risks.

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

# A. Related Work

## A.1. Data Selection for LLMs

Data selection for LLM training can generally be categorized into two approaches: heuristic-based and model-based. Heuristic-based methods, such as text extraction (Barbaresi, 2021), language filtering (Joulin et al., 2017), and data deduplication (Tirumala et al., 2023), are typically the initial stages of pre-processing web-crawled data. Model-based data selection methods can follow these stages to further refine the data by considering various factors, such as quality (Li et al., 2024b), diversity (Liu et al., 2024), and distribution matching (Kang et al., 2024). Model-based selection can be conducted at the sample level (Fan & Jaggi, 2023; Gu et al., 2025), where a complete piece of text is retained or discarded, or at the token level (Lin et al., 2024), where only useful tokens are incorporated into LLM training. These methods are generally targeted at the general domain, often overlooking the specific features of mathematical and reasoning problems. Closest to our work, (Zhang et al., 2024b) proposed using designed prompts for LLMs to evaluate whether a text exhibits mathematical intelligence and if it is suitable for educational purposes in mathematics. In contrast, our work addresses this problem from the perspective of skills.

## A.2. LLMs for Mathematical Reasoning

Mathematical reasoning is a cornerstone for assessing the fundamental cognitive capabilities of human intelligence. Recently, there has been a notable surge in the development of LLMs (Shao et al., 2024; Yang et al., 2024) aimed at solving mathematical reasoning problems. The release of OpenAI's o-series models has further highlighted the potential, significance, and opportunities for enhancing the reasoning capabilities of large models. From the dataset perspective, numerous high-quality datasets have been developed to bolster the mathematical reasoning abilities of LLMs. For instance, in the SFT stage, OpenMathInstruct-2 (Toshniwal et al., 2025) features 14 million question-solution pairs, using various augmentation methods from the MATH (Hendrycks et al., 2021) and GSM8k (Cobbe et al., 2021) datasets, making it nearly eight times larger than the previous largest open-source math reasoning dataset. In the pretraining stage, MathPile offers a diverse and high-quality math-centric corpus comprising about 9.5 billion tokens. This corpus was developed through meticulous data collection and processing, including a comprehensive suite of preprocessing, prefiltering, language identification, cleaning, filtering, and deduplication. Our method can be further utilized to select these existing datasets through the math skill graph, thereby efficiently and effectively training LLMs.

## A.3. Graphs in Data Curation for LLMs

Graphs, as a form of data that encapsulate compositional relationships, can be valuable in analyzing the acquisition of knowledge during the training process of LLMs. Skill-it (Chen et al., 2024) demonstrates that language models follow a natural order when learning a set of skills from their training data and builds a skills graph for data-efficient LLM training. However, the skills they extracted are relatively general and human-designed, not specifically centered on mathematical skills. MathScale (Tang et al., 2024) instructs LLMs to construct a concept graph and uses a graph random walk algorithm to sample a sub-graph, which is then used to generate an instruction tuning dataset. Similarly, Zhou et al. (Zhou et al., 2024b) further demonstrate the effectiveness of graph-based instruction data generation by constructing dataset-related graphs. These two methods differ from ours, which focuses on data selection for the continual pretraining stage of LLMs.

# B. More Information of Skill Graph

## B.1. Visualization of Skill Graph

We show a sampled sub-skill graph with 10 nodes.

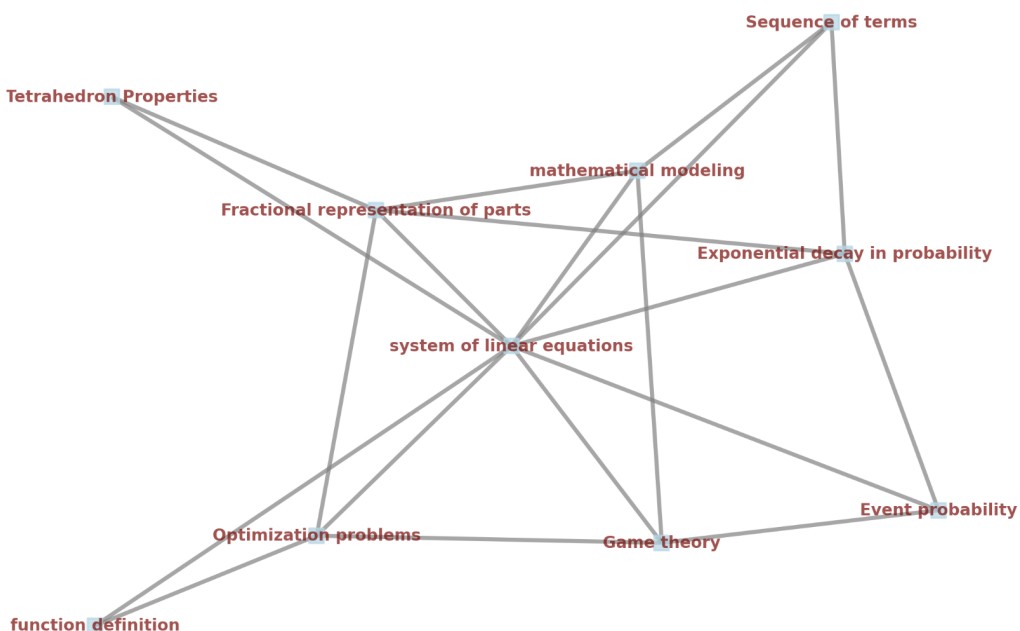

*Figure 6.* A visualization of sub-skill graph.

## B.2. Statistics of Skill Graph

*Table 6.* Skill Graph Properties

| Property | Value |
|---|---|
| Number of nodes | 46,490 |
| Number of edges | 1,230,497 |
| Density | 0.001 |
| Clustering Coefficient | 0.776 |
| Modularity | 0.587 |
| Average degree | 52.94 |
| Maximum degree | 11,691 |
| Minimum degree | 4 |
| Degree standard deviation | 199.69 |

## C. Skill Extraction Details

The prompt used to extract mathematical skills from the referenced dataset is presented below.

---

**Prompt Template to Extract Skills**

Please assume the role of a math teacher and analyze the provided question with the following steps:

1. Determine if the text involves mathematical knowledge, reasoning, or problem-solving skills. Respond with "YES" or "NO".
2. Identify 1-10 concise, general mathematical knowledge points being tested.

Note:
1. Ensure each knowledge point is abstract and generalized.
2. Avoid using verbs; focus on noun phrases (e.g., "function symmetry" instead of "analyzing function symmetry").
3. Avoid specifics like numbers, angles, or exact values. Focus on overarching concepts and techniques.
4. Keep each knowledge point to a maximum of 10 words.

Provide your assessment in one JSON format:

```json
{
"math relevance": "YES" or "NO",
"knowledge points": ["point1", "point2", ...]
}
```

Provided text:
{**TEXT**}

JSON output:

---

## D. Case studies

We show the data samples with the highest and lowest scores in Jiuzhang3.0, OpenWebMath-pro and OpenWebMath datasets, respectively.

---

**Text with the highest score in Jiuzhang3.0 dataset**

### Problem:

Consider a sequence of numbers where each term is the result of applying the function $\overline{x}_5$, which maps any integer $x$ to its smallest nonnegative integer equivalent when divided by 5. For example, $\overline{7}_5 = 2$ because $7 \div 5$ has a remainder of 2.

Let $S(a, r)$ represent the sum of $\overline{(a - \ell)}_5$ for $\ell$ ranging from 0 to $r$, where $a$ and $r$ are integers.

Given $a = 17$ and $r = 12$, calculate the value of $S(17, 12)$.

### Solution

To solve for $S(17, 12)$, we use the formula provided in the proposition:

$$S(a, r) = \begin{cases} a(r + 1) - \frac{r(r+1)}{2} & \text{if } r \leq a \\ a(r + 1 - 5) + 5r - \frac{r(r+1)}{2} & \text{if } r > a \end{cases}$$

Since $r = 12$ and $a = 17$, we see that $r > a$. Therefore, we use the second case of the formula.

First, calculate $a(r + 1 - 5)$:

$17 \cdot (12 + 1 - 5) = 17 \cdot 8$

$17 \cdot 8 = 136$

Next, calculate $5r - \frac{r(r+1)}{2}$:

$5 \cdot 12 - \frac{12 \cdot (12+1)}{2} = 60 - \frac{12 \cdot 13}{2}$

$60 - 78 = -18$

Now, sum the two parts:

$136 + (-18) = 136 - 18 = 118$

Thus, the value of $S(17, 12)$ is:

$\boxed{118}$

### Explanation

In this problem, we needed to apply the formula for the sum of terms involving the function $\overline{a - \ell}_5$. The key step was recognizing which case of the formula to use based on the values of $a$ and $r$.

By breaking down the problem into simpler calculations and carefully handling the algebraic expressions, we were able to logically determine the correct value for the sum. This required a clear understanding of the function properties and the conditions for applying different formula cases. This approach of systematically analyzing the problem using established methods will help students in their mathematical journey.

---

**Text with the lowest score in Jiuzhang3.0 dataset**

Sven Erlander was born in 1934 and died in 2021. How many years old was Sven Erlander when he died?

### Solution

To find out how old Sven Erlander was when he died, we need to follow these steps:

1. **Determine Sven Erlander's birth year:** Sven Erlander was born in 1934.

2. **Determine the year he died:** Sven Erlander died in 2021.

3. **Calculate the age difference:** We subtract his birth year from the year he died:

$$2021 - 1934 = 87 \tag{14}$$

So, Sven Erlander was 87 years old when he died.

## New exercises and problems in MathematicsSeptember 2003

**New exercises**

Maximum score for each exercise (sign "C") is 5 points.

A coin has been placed in each field of a 3x3 table, showing tails on top. At least how many coins need to be turned over, so that there are no three collinear (row, column, diagonal) heads or three collinear tails?

Is there a regular polygon in which the shortest diagonal equals the radius of the circumscribed circle?

Peter's telephone number (without area code) is 312837, that of Paul is 310650. If each of these numbers is divided by the same three-digit number, the remainders will be equal. That remainder is the area code of their city. What is the remainder? (Note: Area codes are two-digit numbers in Hungary.)

The angles A and B of a convex quadrilateral ABCD are equal, and angle C is a right angle. The side AD is perpendicular to the diagonal BD. The lengths of sides BC and CD are equal. What is the ratio between their common length and the length of side AD?

Solve the equation $2x \log x + x - 1 = 0$ on the set of real numbers. (Suggested by É. Gyánó, Budapest)

**New problems**

The maximum scores for problems (sign "B") depend on the difficulty. It is allowed to send solutions for any number of problems, but your score will be computed from the 6 largest scores in each month.

We have coloured each positive integer either red or blue. The sum of two numbers of different colours is always blue, and their product is always red. What colour is the product of two red numbers? (3 points)

Find the locus of those points in the plane of a given square at which the square subtends an angle of 30°. (3 points)

Prove that if $m$ and $n$ are integers, $m^2 + n^2 + m + n - 1$ cannot be divisible by 9. (3 points)

The convex hexagon ABCDEF is cyclic, and $AB = BC = a$, $CD = DE = b$, $EF = FA = c$. Prove that the area of the triangle BDF is half of the area of the hexagon. (4 points)

The number $F$ in base-$a$ notation is $0,3737\ldots = 0,\dot3\dot7$ and the number $G$ in base-$a$ notation is $0,7373\ldots = 0,\dot7\dot3$. The same numbers written in base-$b$ notation are $F = 0,2525\ldots = 0,\dot2\dot5$ and $G = 0,5252\ldots = 0,\dot5\dot2$. Determine the numbers $a$ and $b$. (4 points)

Is there a right-angled triangle such that the radius of the incircle and the radii of the three excircles are four consecutive terms of an arithmetic progression? (4 points)

The point $P$ lies on the perpendicular line segment dropped from the vertex $A$ of the regular tetrahedron $ABCD$ onto the face $BCD$. The lines $PB$, $PC$ and $PD$ are pairwise perpendicular. In what ratio does $P$ divide the perpendicular line segment? (3 points)

Given the real number $t$, write the expression $x^4 + tx^2 + 1$ as a product of two quadratic factors of real coefficients. (4 points)

The points $X$, $Y$ and $Z$ divide a circle into three arcs that subtend angles of 60°, 100° and 200° at the centre of the circle. If $A$, $B$ and $C$ are the vertices of a triangle, let $MA$ and $MB$ denote the intersections of the altitudes drawn from the vertices $A$ and $B$ with the circumscribed circle, and let $FC$ denote the intersection of the bisector of angle $C$ with the circumscribed circle. Determine all the acute triangles $ABC$ for which the points $MA$, $MB$ and $FC$ coincide with the points $X$, $Y$ and $Z$ in some order. (4 points)

Let $x_1 = 1$, $y_1 = 2$, $z_1 = 3$, and let $x_{n+1} = y_n + \frac{1}{z_n}$, $y_{n+1} = z_n + \frac{1}{x_n}$, $z_{n+1} = x_n + \frac{1}{y_n}$ for every positive integer $n$. Prove that at least one of the numbers $x_{200}$, $y_{200}$ and $z_{200}$ is greater than 20. (5 points)

**New advanced problems**

Maximum score for each advanced problem (sign "A") is 5 points.

$I$ is the isogonic point of a triangle $ABC$ (the point in the interior of the triangle for which $\angle AIB = \angle BIC = \angle CIA = 120°$). Prove that the Euler lines of the triangles $ABI$, $BCI$ and $CAI$ are concurrent.

Prove that if $a, b, c$ are positive real numbers then

$$\frac{1}{a(1+b)} + \frac{1}{b(1+c)} + \frac{1}{c(1+a)} \geq \frac{3}{1+abc}.$$

We have selected a few 4-element subsets of an $n$-element set $A$, such that any two sets of four elements selected have at most two elements in common. Prove that there exists a subset of $A$ that has at least $\sqrt[3]{6n}$ elements and does not contain any of the selected 4-tuples as a subset.

**Article Content**

**Leo McKern**

Reginald Leo McKern (March 16, 1920 - July 23, 2002), better known simply as Leo McKern, was an Australian actor who appeared in numerous British television programs, movies and in over 200 stage roles.

**Some notable roles:**

**Text with the highest score in OpenWebMath dataset - Part 1**

## Question List

For all nonzero integers $l$ and $m$, let the operation § be defined by

$$lm = -\left|\frac{1+m}{l}m\right|$$

**Quantity A**

$$3\frac{3}{2}$$

**Quantity B**

$$-1$$

There are 30 students in Mr. Peterson's gym class. 14 of them play basketball, 13 play baseball, and 9 play neither basketball nor baseball.

**Quantity A**

The number of students who play both basketball and baseball.

**Quantity B**

$$6$$

In a regular $n$-sided polygon, the measure of a single angle is

$$\frac{(n-2)180}{n}$$

The degree measure of an angle in a regular 10-sided polygon is how much greater than the degree measure of an angle in a regular 6-sided polygon?

For all real numbers $a$ and $b$, the operation $\oplus$ is defined by

$$a \oplus b = 2a - b$$

What is the absolute value of the difference between $(3 \oplus 1) \oplus 2$ and $6 \oplus 3$?

**Quantity A**

The units digit of $7^{29}$.

**Quantity B**

The units digit of $3^{27}$.

Of the employees at a company, 60 percent were men and, of these, $\frac{1}{10}$ were still employed after a recent corporate restructuring. If the number of women who were still employed after the restructuring was five times the number of men who were employed after it, what percent of the women were still employed after the restructuring?

If $q$ is even, then $\#q = -2$; If $q$ is odd, then $\#q = -4$. $a$ and $b$ are integers such that $b - 3$ is odd.

$$\#(6a)$$

**Quantity B**

$$\#b$$

$$f(x) = 3x^2$$
$$g(x) = x + 1$$

$x$ is an integer such that $-10 \leq x \leq -1$.

$$f(g(x))$$

**Quantity B**

$$g(f(x))$$

Three digits have been removed from each of the following numbers. If $n = 25$, which of the numbers is equal to $3 \cdot 2^{(n-1)}$?

$$m \parallel n$$

$$a$$

$$90$$

$$r$$

**Quantity B**

$$s$$

If $X$ is the center of the circle above, then what is the sum of the measures of $\angle WXY$ and $\angle VXZ$?

In the figure above, $c$ is $\frac{4}{5}$ of $d$. What is the value of $c$?

What is the value of $x$?

$$a + b$$

**Quantity B**

$$180 - c$$

In the figure above, what is the value of $w$?

What is the area of a regular hexagon with side length 8?

$$a \parallel b$$

$$95$$

**Quantity B**

$$s$$

In the figure above, line $j$ is parallel to line $k$. If $f = 130$ and $g = 70$, then $h =$

$$1\ 2\ \ldots\ 10\ 11\ 12\ 13\ 14\ 15\ 16\ \ldots\ 36\ 37$$

Text with the lowest score in OpenWebMath dataset - Part 1

## M A S H (television)

Inspired by the film of the same name, MH (Mobile Army Surgical Hospital) was an American television series that aired on CBS from September 17, 1972 to February 28, 1983 (251 episodes). The sitcom was about an outfit of medical workers stationed in Korea during the Korean War. Much like the movie, it combined elements of a "zany" comedy and a darker antiwar message.

20th Century Fox head William Self gave the show to producer Gene Reynolds and comedy writer Larry Gelbart. This combining of genres was unusual for television series of its time, and was in fact an early example of what later became known as a dramed. The show's producers did not even want a laugh track in the show, but this proposal was rejected by CBS; however, as a compromise, the emergency room scenes were shown without a laugh track, and in fact the show was shown in the United Kingdom entirely without the use of canned laughter. The DVD release offers a choice between laugh-encrusted and laugh-free soundtracks. Many of the stories were based on real life tales told by hundreds of actual surgeons interviewed for the show.

At the end of its first season the show ended 46th in the ratings. CBS responded by moving the show to Saturday night between hits All in the Family and The Mary Tyler Moore Show. MH ended the next 9 of 10 seasons in the top 10.

The series used the theme song "Suicide is Painless", which was taken from the film, though without the lyrics. Some said the series seemed to be more about the Vietnam War, given the attitudes of the characters, than about the Korean War – despite its Korean setting. However, even the movie was somewhat anachronistic, given its use of such early seventies fashion as the fu manchu mustache. The show's producers have said that the movie was really about all wars, not just Korea or Vietnam.

The series was followed by After starting Morgan, Farr, and Christopher reunited in a midwestern hospital after the war. It was not well regarded, and was quickly cancelled.

The show featured Alan Alda, who wrote and directed some of the most emotional and award-winning episodes; Out of all the starring characters Hawkeye, Hotlips, Klinger and Father Mulcahy were the only ones in the show for its entire series. McLean Stevenson left the show at the end of the third series, and his character Henry Blake was discharged and sent home. In the final scene of his last episode it was reported that Blake's plane had been shot down and he was killed. Actor Wayne Rogers left the series after the end of series three due to disagreements about his character. At the beginning of the fourth series Hawkeye was informed by Radar that Trapper had been discharged, and audiences did not see Trapper's departure. At the same time Col Potter was assigned to the unit as Commanding Officer replacing Blake, while BJ Hunnicut was drafted in as Trapper's replacement. Larry Linville left during the first episode of series six as Frank Burns became mad and was drafted away from the 4077th. Charles Winchester, a snobbish but highly skilled surgeon, was his replacement. A couple of episodes into series eight Gary Burghoff left the series, and Radar was discharged. Existing character Klinger took over Radar's post, the character thereafter enjoying a more prominent position in the series.

Gary Burghoff (Radar O'Reilly) was the only M$^*$A$^*$S$^*$H actor to reprise his role from the movie, retaining his extraordinary ability to detect the arrival of choppers transporting wounded long before anyone else could hear a thing. When Burghoff left the series, the company clerk role was taken up by Jamie Farr (Corporal (later Sergeant) Klinger, whose cross-dressing never got him the discharge he wanted).

Viewed as one of the most popular sitcoms in history, it is still a very popular syndicated series. Originally seen as an ensemble show, it became increasingly centered around Alan Alda's character, Hawkeye Pierce. The show survived many personnel changes over the course of the show, and in fact the series changed its tone over the years. Initially, the series placed more emphasis on the "zany" elements, while in the later years it focused more on serious elements and character development; however, both the serious and the comedic elements were present throughout the history of the series. In the later years the story lines began to also stale and the show's comedic edge had dulled even though the show was still in the top of the ratings. Alda and his fellow actors then voted to end the series with the 10th season but CBS and 20th Century Fox offered the actors a shortened 11th season leading up to an opportunity for them to say goodbye in a grand finale.

Text with the lowest score in OpenWebMath dataset - Part 2

Goodbye, Farewell, and Amen The final episode was titled "Goodbye, Farewell, and Amen" and was first broadcasted on February 28, 1983. The episode was 2.5 hours long and was viewed by over 125 million Americans (77% of viewership that night) which made "Goodbye, Farewell, and Amen" the most watched television episode in history up to that time.

The finale started in the waning days of the war with Hawkeye in a mental hospital who was finally driven over the edge by a bus ride gone terribly wrong. The bus passengers, who were refugees, were in danger of being discovered and executed by a North Korean patrol. Hawkeye scolds the refugees to be quiet but a baby begins to whimper and its mother responds by smothering the child. Hawkeye repressed this by replacing the memory of the baby with that of a small animal.

Dr. Winchester captured a rag-tag bunch of Chinese musicians who he teaches Wolfgang Amadeus Mozart's "Quintet for Clarinet and Strings" to them. However, he later sees all the musicians killed and as a result views classical music as stained for him (classical music was his number one solace during the war).

Corporal Max Klinger, best known for constantly trying to be discharged via a Section 8, finally decides to stay in Korea to be with his new wife even though he finally got his release papers (along with most of the 4077).

The final and perhaps most memorable scene was between Hawkeye and BJ Hunnicut. Hunnicut was not able to say goodbye and Hawkeye mocked him for this failure. Both men lament that they will be on opposite sides of the country after they go home and conclude that they will probably never see each other again. They tearfully embrace for the last time and Hawkeye boards a helicopter and lifts off. Hunnicut rides off on a motorcycle and as the helicopter ascends Hawkeye sees a final message from his longtime friend spelt out with stones on the sandy soil, "GOODBYE."

