# OpenReview forum: "MASS: Mathematical Data Selection via Skill Graphs for Pretraining Large Language Models"
_ICML.cc/2025/Conference — ICML 2025 poster_

### Official Review · Reviewer_2U9L · 2025-03-13

**Overall Recommendation:** 4

**Summary:**

The paper proposed MASS, a novel mathematical skill graph construction method for selecting data for pretraining LLMs in the math domain. MASS prompts a strong LLM to generate nodes of skills from a reference dataset, and then construct an adjacency matrix (as a graph) using the dataset statistics. Then the graph can be leveraged to calculate quality scores for the pretraining data by their representational similarity with the skills. Empirically, MASS outperforms other pretraining data selection baselines and full-dataset training. More broadly, MASS demonstrates the great potential of using graphs for data selection in general.


## update after rebuttal
I have increased my score as my concerns are well addressed by the authors.

**Claims And Evidence:**

Most of the claims are well-supported.

**Essential References Not Discussed:**

N/A.

**Experimental Designs Or Analyses:**

The experimental designs for evaluation and ablation studies are comprehensive. However, more baselines (e.g., DSIR) could be included for pretraining data selection on Mistral-7B.

**Methods And Evaluation Criteria:**

The method is brilliant and the evaluation criteria are properly chosen.

**Other Comments Or Suggestions:**

N/A

**Other Strengths And Weaknesses:**

**Strengths**

1. The paper is well-written and well-structured.
2. The methodology is novel and interesting. Using the adjacency matrix of the skill graph to select pretraining data not only accounts for their direct usefulness to learn individual mathematical skills, but also accounts for their potential usefulness to other skills when multiple skills are required to solve a problem.
3. The ablation studies are well executed to demonstrate the performance gain from the graph construction.

**Weaknesses**

1. Although the design of the method, the empirical performance, and the ablation studies are commendable, there is a lack of theoretical understanding on why the proposed approach works better than other baselines.
2. The computational complexities of constructing the skill graph and selecting from the skill graph remain unclear.
3. The comparison with other baselines is only performed on TinyLlama-1.1B. It will be great to show at least a comparison with DSIR on the larger Mistral-7B.

**Questions For Authors:**

1. In line 323 (right column), the author said “for Jiuzhang 3.0, a higher selection ratio leads to better performance”, but when the ratio is too high (>70%), the performance decreases. Could you please clarify the correctness of the sentence? In addition, could you please explain what could be the reasons that caused the spike?
2. Could you provide a bit more statistics about the graph to understand how dense is the graph, e.g., clustering coefficient?
3. In equations (1) and (2), the entries are normalized with different normalizing factors so that they can sum up to 1 respectively. Why is the normalization important?

**Relation To Broader Scientific Literature:**

Selecting pretraining data using a carefully constructed graph that contains knowledge and statistics is a novel and brilliant idea, which is under-explored by prior works according to my knowledge. Although this paper only demonstrated its practical usability in the domain of mathematics, I believe it has a great potential for data selection for other domains in general.

**Theoretical Claims:**

N/A.

---

> ### Author Rebuttal · Authors · 2025-04-01
>
> Dear Reviewer 2U9L,
>
> Thank you for your thoughtful feedback and positive recognition. Below, we respond to each of your comments in detail.
>
> 1. **Weakness 1:** There is a lack of **theoretical understanding** on why the proposed approach works better than other baselines.
>
>     **A:** First, we emphasize that this work prioritizes model architecture and performance improvements over rigorous theoretical analysis, which we defer to future research.
>
>     Second, we present some initial thoughts and general insights. In Section 2.4, we analyze why MASS works through the lens of skills. Here, we highlight two features of the MASS-selected data:
>     - It encompasses a wider range of important mathematical skills.
>     - It captures a richer, more nuanced compositional understanding of these skills.
>
>     In contrast, other baselines adopt fundamentally different approaches:
>     - Rule-based method focuses more on pre-processing, such as language filtering and deduplication.
>     - Rho-1 improves the data quality within each sample at token-level.
>     - ProX, following Rho-1, uses programming to refine data.
>     - DSIR calculates the weight of each sample using bag-of-ngrams estimator.
>     - AutoDS prompts an LLM to select data simply based on its general math intelligence.
>
>     We believe what seperates MASS from other baselines is that we select data at a finer semantic granularity—skills. Having the reference data, we decide what skills models really need and lack, so we can purposely select a subset that contains these skills to teach LLMs. Other baselines, meanwhile, prioritize linguistic-level quality (e.g., grammar, noise, or coherence) without considering the sementics and skill-level efficacy.
>
> 2. **Weakness 2:** The **computational complexities** of constructing the skill graph and selecting from the skill graph remain unclear.
>
>     **A:** Here we break down the computational complexities of the whole approach of MASS.
>
>     | Operations  | A100 GPU hours |
>     |--------|------|
>     |Extracting skills from reference dataset|~24|
>     |Embedding reference dataset and target dataset|~0.5|
>     |Constructing skill graph|~2 CPU hours|
>     |Selecting high-quality subset from target dataset|~4|
>     |Training Mistral-7B on high-quality subset (~10B tokens)|~960|
>
>     As shown, the computational cost of pre-processing steps is relatively low compared to model training (<3%).
>
> 3. **Weakness 3:** Show **a comparison with DSIR on the larger Mistral-7B**.
>
>     **A:** Thank you for your suggestion. We began training Mistral-7B using DSIR, and the model is still in training. We will post the results in the discussion phase as soon as complete.
>
> 4. **Question 1:** Clarify the **correctness** of line 323 / Explain what could **cause the spike**.
>
>     **A:** We confirm this statement is correct. 'Higher' is used when we compare Jiuzhang3.0 and OMW-pro, not with Jiuzhang3.0. In figure 5, it is clear that the best ratio for Jiuzhang is higher than that for OWM-pro.
>
>     As for the spikes, we offer the following explanation:
>     - At very low selection ratios, while the selected data maintains high quality, the reduced diversity negatively impacts model performance. Prior research [1] has demonstrated the importance of data diversity in selection methods.
>     - When the selection ratio is too high, too much noisy and low quality data is remained thus harming the performance.
>
>     Thus, the observed spike-shaped relationship between selection ratio and model performance is well-justified from both quality and diversity perspectives as a trade-off.
>
>
> 5. **Question 2:** More **statistics about the graph**.
>     | Property   | Value|
>     |--------|------|
>     | Number of nodes   | 46,490  |
>     | Number of edges   | 1,230,497 |
>     | Density  | 0.001 |
>     | Clustering Coefficient | 0.776 |
>     | Modularity| 0.587 |
>     | Average degree | 52.94 |
>     | Maximum degree | 11691 |
>     | Minimum degree | 4 |
>     | Degree standard deviation | 199.69|
>
> 6. **Question 3:** Why is the **normalization in Equ. 1 and 2** important?
>
>     **A:** We normalize the entries in equations (1) and (2) with different temperature coefficients because these original entries (raw counts of skills and co-occurrences) follow different statistical distributions. This normalization ensures both measures are on comparable scales before being combined in Equation (4) to compute similarity. Without such scaling, one type of entry could disproportionately influence the final similarity measure due to its inherently larger magnitude. The temperature-based normalization prevents either component from dominating the results while preserving their relative importance.
>
> Thank you again for your thoughtful advice. We will include relevant details and analysis in the next version of our manuscript.
>
> Sincerely,
>
> MASS authors
>
> [1] Harnessing Diversity for Important Data Selection in Pretraining Large Language Models, ICLR 2025.

---

> > ### Comment · Reviewer_2U9L · 2025-04-02
> >
> > Thank you for the detailed comments. The authors have addressed most of my concerns well. However, I still don't understand why line 323 is correct (question 1):
> >
> > > We confirm this statement is correct. 'Higher' is used when we compare Jiuzhang3.0 and OMW-pro, not with Jiuzhang3.0. In figure 5, it is clear that the best ratio for Jiuzhang is higher than that for OWM-pro.
> >
> > Line 323 says "a higher selection ratio leads to better performance". While the best-performing ratio of Jiuzhang3.0 is higher than that for OWM-pro, it is not clear that this justifies "higher selection ratio leads to better performance". What performance is being compared, and whose performance is being evaluated? Please clarify.
> >
> > ---
> >
> > I have read comments from other reviewers. I do not have other concerns about the paper. However, I find the questions from Reviewer CEvW quite interesting, especially Q1 regarding the understanding of the compositionality knowledge from equation 10, which I don't think the authors have addressed in response to Weakness 1 in my review. It would be great to hear more thoughts from the authors.

---

> > > ### Author Response · Authors · 2025-04-02
> > >
> > > ## **Update:** The experimental results for additional baselines are now available. We have included DSIR and BM25 for pretraining data selection on Mistral-7B.
> > >
> > > For BM25 method, we implement it using the repo [3]. We have randomly selected 100 samples from the reference dataset and ranked the target dataset based on the representation to select a high-quality subset. Also, we include DSIR baseline by using their official repo as reviewer 2U9L suggested. We continued pretraining Mistral-7B using variants of OpenWebMath-pro for ~5B tokens:
> > >
> > > |Data|asdiv|gsm8k|mathqa|mawps|minerva_math|mmlu_stem|svamp|tabmwp|Avg.|
> > > |-|-|-|-|-|-|-|-|-|-|
> > > |Orig.| 73.7|47.1|42.6|89.5|21.8|52.2|63.2|58.2|56.1|
> > > |BM25|73|44.7|49.8|86.1|24|52.6|63.1|49.1|55.3|
> > > |DSIR|73.4|42.1|55.3|86.8|21.6|51.9|63.6|50.4|55.6|
> > > |MASS|76.8|53.2|51.8|90.4|25.6|54.5|67|57.6|59.6|
> > >
> > > MASS still outperforms other baselines by at least 3%. Surprisingly, BM25 and DSIR perform worse than using the original data. We hypothesize that this is due to their limited diversity, as BM25 and DSIR score data points based on linguistic-level features (e.g., BM25 and n-gram similarity between the reference and target data).
> > >
> > > ---
> > >
> > > Thank you for your quick reply. As for line 323, in your reply, you mentioned that:
> > > > While the best-performing ratio of Jiuzhang3.0 is higher than that for OWM-pro.
> > >
> > > This is exactly what we meant. Or we could rephrase it as
> > > > For Jiuzhang3.0, a relatively high selection ratio leads to high performance. For OpenWebMath-pro, a relatively low selection ratio leads to high performance.
> > >
> > > We wrote this because we wanted to demonstrate that a dataset with higher quality probably matches a higher selection ratio, like Jiuzhang3.0.
> > >
> > > Now we have realized the original sentence might seem a bit confusing and we will refine it in the next revision.
> > >
> > > ---
> > > Thank you for reading other reviews as well. We planed to reply to reviewer CEvW's question in the discussion phase because of the space constraints. Here we would like to put our thoughts:
> > >
> > > **A:** To analyze Equ. 10, we first need to address Equ. 4 which calculates the aggregated similarity of a data point $x_i$ and a skill $v_j$. The first term is straightforward: it yields a high score when both the skill frequency ($A_{j,j}$) and the original similarity ($\mathrm{sim}(x_i, v_j)$) are high. The second term captures co-occurrence patterns: when skills $v_j$ and $v_k$ frequently co-occur ($A_{j,k}$ is high) and the point $x_i$ is also similar to $v_k$ ($\mathrm{sim}(x_i, v_k)$ is high), this contributes to the aggregated similarity.
> > >
> > > So the fact that the relationship of skill $v_j$ and its relevant skill $v_k$ affects the importance score of a sample $x_i$ is the reason why we claim MASS contains compositional knowledge. Although it feels that we are scoring the sample based on a coarser grained skill/"skill family" as reviewer commented, we specifically characterize this as a compositional feature rather than a robustness mechanism. Finally, by summing over all skills in Equ. 4, we obtain Equ. 10, which gives the final importance score for $x_i$.
> > >
> > > As for your version of the second term in Equ. 10, it can be rewriten as $\mathrm{sim}(x, v_j)\sum_{v_k \in \mathcal{N}(v_j)}\mathbf{A}_{j,k}\mathrm{sim}(x, v_k)$ where the extra term $\mathrm{sim}(x, v_j)$ can be seen as a constant coefficient used to regulate the second term, but in fact we have already taken $\mathrm{sim}(x, v_j)$ into acount in the first term of Equ. 10. As a result, we believe that both our original version and reviewer's version share the same philosophical principles behind and reflect the compositional knowledge we claim, while they may have slightly different forms and numerical values. We would love to explore reviewer's implementation in future work.
> > >
> > > As for enforcing a looser skill taxonomy and only using the first term, we have done it in our ablation study-impact of skill graph-w/o non-diagonal entries section. This variant is equivalent to only using the first term in Equ. 10 and we found that it does degrade the performance by at least 2%. This empirical result substantiates our claim that MASS effectively captures compositional knowledge through its complete formulation.

---

### Official Review · Reviewer_CEvW · 2025-03-14

**Overall Recommendation:** 3

**Summary:**

This paper proposes an approach called MASS for selecting mathematical training data. The paper takes a high-quality reference math dataset, obtains each problem's skills (by prompting a LM), and constructs a skills graph, from which we can read off how frequent each skill is in the reference dataset and which skills commonly occur together. MASS uses the skills graph to select samples from a target dataset, prioritizing samples that are associated with the most commonly occurring skills in the reference dataset, and samples that have compositional knowledge. Experiments show that continually pretraining with MASS outperforms other common data selection methods.

**Claims And Evidence:**

I have no concerns about the claims made regarding whether MASS works; however, I do have some questions about the claims regarding *why* MASS works, and would appreciate any clarification. The paper claims that MASS encourages selecting samples that cover compositional information, but I am not fully convinced that this is the entire explanation of what is actually happening.

When looking at the first term in equation 10, my interpretation of MASS is that it uses the frequency of skills, $[A_{11}, A_{22}, ...,]$ as a representation of a reference dataset. With just the first term, the score function encourages selecting points whose associated skills (as captured by $sim(x, v_j)$) are those that frequently occur in the reference dataset; i.e., we select target dataset samples that are similar to the reference dataset samples in terms of the skills they exhibit. The second term in the scoring function appears to cover a neighborhood of v_j, which feels like its a robustness mechanism, rather than something that explicitly encourages compositionality. That is,using $\mathcal{N}(v_j)$ in addition to $v_j$ feels like we are scoring the sample based on a coarser grained skill/"skill family" in the skills graph, which provides robustness against if individual skills are too narrow/poorly defined. Mathematically, the point I am trying to get across becomes clearer if we rewrite equation 10 slightly: $score(x) = \sum_{j=1}^{|V|} \sum_{v_k \in v_j \cup \mathcal{N}(v_j)} A_{jk} sim(x, v_k)$. For each skill $v_j$, you are summing over the skills graph entries corresponding to its neighborhood (including itself).

To further understand why MASS works, I have two questions/thoughts:
- For equation 10, what if the second term was $A_{jk} sim(x, v_k) sim(x, v_j)$? That is, we'd increase the score of $x$ a lot if it was similar to both $v_k$ and $v_j$, and $v_k$ and $v_j$ tend to occur together a lot in the reference dataset. This feels to me like it would capture more of this compositional knowledge you claim.
- It would be interesting to investigate if enforcing a looser skill taxonomy and only using the first term would essentially result in the same sort of data being selected.

**Essential References Not Discussed:**

None.

**Experimental Designs Or Analyses:**

The experimental setup appears well-structured, although I had a few concerns:
- Compute efficiency analysis: how do these data selection methods perform if we control for compute in the data selection process? It would be interesting to compare the performance of MASS to the performance of allocating the compute used for MASS's data selection to just train on more random data from the target dataset.
- AutoDS baseline: do you prompt using the same LM for AutoDS and MASS? If I remember correctly, AutoDS uses Qwen-72B base language model while MASS uses Qwen2.5-72B-Instruct-GPTQ-Int4. More details on implementation of baselines would be appreciated.
- Additional baselines: additional baselines that select data based on matching some representation of a reference dataset, like LESS https://arxiv.org/abs/2402.04333 or BM25, could be helpful.

**Methods And Evaluation Criteria:**

See "Claims and Evidence" above. The evaluation criteria makes sense, and the proposed method makes sense overall, besides the following two points (rephrased from above):
- For equation 10, what if the second term was $A_{jk} sim(x, v_k) sim(x, v_j)$? That is, we'd increase the score of $x$ a lot if it was similar to both $v_k$ and $v_j$, and $v_k$ and $v_j$ tend to occur together a lot in the reference dataset.
- I see the appeal of making the LM identify skills in an unsupervised manner---you don't require much domain knowledge. However, I wonder if making the LM list 10 skills per sample results in hallucinations/poorly defined skills. That is, is there some benefit to defining a set of skills/level of granularity (for example, skill = the name and level of the math class you would learn about this problem in), and getting the LM to adhere to this taxonomy?

**Other Comments Or Suggestions:**

None.

**Other Strengths And Weaknesses:**

Strengths:
- Doing data selection by extracting the skills from a reference data and selecting the target samples that are most aligned with these skills is a very interesting and novel idea. It provides evidence towards this skills-based view of how models learn from data, which is both scientifically and practically important.
- The method performs very well and outperforms other data selection approaches.
- The paper is generally well-written and easy to read.

Weaknesses:
- It is not completely clear when this approach works; the paper is lacking recommendations for how to use this approach in practical settings.
     - Does this approach work for other domains?
     - Does this approach work when a practitioner wants to define their own set of skills? What sort of skills work the best for this method? (i.e., topic-based, reasoning-based, style-based).
     - Does the choice of LM matter for extracting skills?
- It is also not completely clear why this approach works; see my comments above regarding my interpretation of the score as encouraging fuzzy neighborhood-based skill alignment, rather than compositional knowledge.
- There should be more information on how baselines were implemented and why these baselines were chosen. Moreover, I don't think there is a clear intuitive reason stated regarding why MASS should outperform these other approaches; why are skills a better axis for data selection than, i.e., directly prompting an LM to score things or a token-level data selection approach?

**Questions For Authors:**

1. Understanding the claim of encouraging compositional information: For equation 10, what if the second term was $A_{jk} sim(x, v_k) sim(x, v_j)$?
2. What is the impact of prompting for different skills (i.e., a pre-defined skills taxonomy, or having the model output 5 or 20 skills rather than 10)?
3. How does MASS compare against other approaches when we match the amount of end-to-end compute used to train the model?
4. Clarification: do you prompt using the same LM for AutoDS and MASS?
5. Can MASS be applied to other domains, like code, science, or natural language?
6. What types of skills work well with MASS, and what skills do not?
7. Why were these baselines chosen and why should one expect MASS to intuitively outperform them?

**Relation To Broader Scientific Literature:**

The contributions are highly relevant to broader scientific literature, since data selection is a critical part of the LM development pipeline. The idea of skills as a way to capture different characteristics of data and matching along skill distributions is interesting and builds on important lines of work that examine how models learn from data. It provides an alternative to existing, less interpretable data selection algorithms, which involve embedding the reference dataset, training on it, computing gradients on it, and so on.
I do think the paper would be more interesting/stronger if the authors investigated if skills graphs could be applied to other domains, such as code, science, or general natural language.

**Theoretical Claims:**

Not applicable, no theorems in paper.

---

> ### Author Rebuttal · Authors · 2025-04-01
>
> Dear Reviewer CEvW,
>
> Thank you for your thoughtful feedback and positive recognition. Below, we respond to each of your comments in detail.
>
> 1. **Weakness 1 Q1 / Question 5:** Does this approach work for **other domains**?
>
>     **A:** Yes, it does work for other domains. Please see our reply to reviewer NxxW on Question 1.
> 3. **Weakness 1 Q2 / Question 6:** Does this approach work when a practitioner wants to define their **own set of skills?** What sort of **skills work the best** for this method? (i.e., topic-based, reasoning-based, style-based).
>
>     **A:** For the first question, yes, it does work with minor adjustments. We propose two potential solutions:
>     - We can directly prompt LLMs to identify relevant skills from a pre-defined set for each math sample. However, this faces challenges when the skill set is large (e.g., our 46,490 distinct skills) due to: (1) prompt length constraint and (2) model's instruction following ability.
>     - We can opt for a BERT-like classfier model instead of autoregressive models. So we need to collect a sufficiently large training set of math data and their corresponding skills pairs to train the BERT model which can further be used at scale for pre-training corpus. The data collection can be done either by utilizing LLMs or manual annotation by experts.
>
>     For the second question, while our current version doesn’t specify skill types during extraction, this might be a direction worth trying. We believe that hierachical topic-based skills may work well. Specifically, we first identify math data point to algebra, geometry, statistics and so on. Next, we further classify algebra data to abstract algebra, lie algebra and so on at a finer level. Thus it also incorporates flexibity and hierachical knowledge.
>
>     However, this remains speculative and what works the best still needs comprehensive designs and experiments to figure out. We leave this for future work.
> 5. **Weakness 1 Q3:** Does the **choice of LM** matter for extracting skills?
>
>     **A:** During our initial implementation, we compared the SOTA proprietary model GPT-4o and open-sourced Qwen2.5-72B-Instruct-GPTQ-Int4 and found that the extracted skills are similar. Using the same example in our manuscirpt, here we show the outputs:
>
>     *GPT-4o:* ["rational expressions","factoring polynomials","root identification","solving rational equations","quadratic equations","expanding algebraic expressions","solving equation","excluding restricted values","considering inequality"]
>     *Qwen2.5:* ["Equation solving", "Factoring polynomials", "Fraction manipulation", "Quadratic equations", "Root identification", "Expression simplification", "Algebraic transformation", "Polynomial division", "Inequality consideration", "Solution verification"]
>
>     Actually what matters more is the prompt template and the output parsing process. After careful evaluation, we selected Qwen2.5-72B-Instruct-GPTQ-Int4 for its optimal balance between cost and performance. This choice allowed us to focus resources on extensive prompt engineering, which ultimately contributed to MASS's significant performance improvements.
>
>     While the current results are satisfactory, we acknowledge that skill extraction could potentially benefit from stronger LLMs and prompt template.
> 8. **Weakness 3 Q2 / Weakness 3 Q3 / Question 7:** A clear intuitive reason regarding **why MASS should outperform these other approaches**; **why are skills a better axis** for data selection than, i.e., directly prompting an LM to score things or a token-level data selection approach?
>
>     **A:** Please see our reply to reviewer 2U9L on Weakness 1.
> 11. **Question 2:** What is the **impact of prompting for different skills** (i.e., a pre-defined skills taxonomy, or having the model output 5 or 20 skills rather than 10)?
>
>     **A:** For a pre-defined skills taxonomy method, please see our reply to Weakness 1 Q2.
>
>     For the number of skills extracted, we need to emphasize that our prompt asks the LLM to output 1-10 skills (not a fixed 10), so the ouput actually varies based on the sample. Moreover, the core contribution of our approach resides in its skill-based framework rather than in quantitative aspects of skill extraction, so we chose not to focus extensively on determining the exact number of skills.
>
> ## Due to space constraints, we are unable to address all questions in this section. We will provide other responses during the discussion phase. Thanks for your understanding.

---

> > ### Comment · Reviewer_CEvW · 2025-04-04
> >
> > Thanks for your response. I also looked through your response to reviewer 2U9L regarding the form and interpretation of equation 10; to me it still intuitively reads more as handling neighborhoods of skills, but I think both interpretations are fine. I will keep my score for now, but look forward to the results from currently ongoing experiments.

---

> > > ### Author Response · Authors · 2025-04-05
> > >
> > > Thank you for your feedback. Below, we address the remaining questions and present the experimental results.
> > > 1. **Weakness 3 Q1 / Concern 2 / Question 7:** How baselines were implemented and why these baselines were chosen.
> > >
> > >     **A:** The reason why we chose these baselines is that we researched on LLM data selction literature, and we tried to include at least one baseline from each catogory. So we have RULE: rule-based method, Rho-1 and ProX: token-level based method, DSIR: n gram feature based doc-level method and AutoDS: LLM based math specific doc-level method.
> > >
> > >     As for how they were implemented, we use the results of RULE and Rho-1 from paper [1]. We implement ProX by using their refined dataset form [2]. We implement DSIR from their official repo and use the default setting to select data. We implement AutoDS from their official repo and Qwen2.5 to select data.
> > >
> > >     For all methods, we trained models using identical hyperparameters (as specified in Table 1 of our manuscript) on their respective selected datasets to ensure fair comparison.
> > > 13. **Question 3 / Concern 1:** How does MASS compare against other approaches when we **match the amount of end-to-end compute** used?
> > >
> > >     **A:** For the compute analysis, please see our reply to reviewer 2U9L on Weakness 2.
> > >
> > >     As shown, the cost of pre-processing steps is relatively low compared to model training (<3%). For other baselines, either we do not know their pre-processing compute or they only require CPU hours so it is hard to precisely match the end-to-end compute.
> > >
> > >     In Figure 3 of our paper, it is clear that MASS achieves ≥40% greater efficiency than all baselines, which means even if we do not consider the pre-processing steps of MASS, it remains the most efficient and effective approach.
> > > 15. **Question 4 / Concern 2:** Do you prompt using the same LM for AutoDS and MASS?
> > >
> > >     **A:** Yes, we use the same model, Qwen2.5-72B-Instruct-GPTQ-Int4 to ensure a fair comparison. Additionally, the Qwen-72B base model is somewhat outdated, and its large size is impractical given the scale of the corpus. In AutoDS, the filtering procedure took ~3,000 A100 GPU hours for 11.26M docs, so we chose a 4-bit quantized yet better-performing model.
> > > ---
> > >
> > > 4. **Concern 3:** **Additional baselines** that select data based on matching some representation of a reference dataset, such as LESS and BM25.
> > >
> > >     **A:** We initially considered LESS as a baseline but later found it unsuitable for our setting. As shown in Table 4 of the LESS paper, their method requires 54 GPU hours for the entire data selection process. However, LESS operates at the instruction-tuning scale (with a maximum dataset size of 18,721 samples) due to the computational cost of gradient-based feature extraction. In contrast, our work focuses on pre-training-scale data (OpenWebMath contains 6.32 million documents), making LESS impractical for comparison. Thus, we excluded it from our analysis.
> > >
> > >     For BM25 method, we implement it using the repo [3]. We have randomly selected 100 samples from reference dataset and ranked the target dataset based on the representation to select high-quality subset. Also we include DSIR baseline by using their offical repo as reviewer 2U9L suggested. We continued pretraining Mistral-7B using variants of OpenWebMath-pro for ~5B tokens:
> > >
> > >     |Data|asdiv|gsm8k|mathqa|mawps|minerva_math|mmlu_stem|svamp|tabmwp|Avg.|
> > >     |-|-|-|-|-|-|-|-|-|-|
> > >     |Orig.| 73.7|47.1|42.6|89.5|21.8|52.2|63.2|58.2|56.1|
> > >     |BM25|73|44.7|49.8|86.1|24|52.6|63.1|49.1|55.3|
> > >     |DSIR|73.4|42.1|55.3|86.8|21.6|51.9|63.6|50.4|55.6|
> > >     |MASS|76.8|53.2|51.8|90.4|25.6|54.5|67|57.6|59.6|
> > >
> > >     MASS still outperforms other baselines by at least 3%. Surprisingly, BM25 and DSIR perform worse than using the original data. We hypothesize that this is due to their limited diversity, as BM25 and DSIR score data points based on linguistic-level features (e.g., BM25 and n-gram similarity between the reference and target data).
> > >
> > > 5. **Stronger base models:**
> > >
> > >     **A:** As reviewer NxxW suggested, we trained a stronger base model, Qwen2.5-7B, using variants of OpenWebMath for ~9B tokens:
> > >
> > >     |Data|asdiv|gsm8k|mathqa|mawps|minerva_math|mmlu_stem|svamp|tabmwp|Avg.|
> > >     |-|-|-|-|-|-|-|-|-|-|
> > >     |-|93.1|85.8|80.6|97.9|53.4|67.5|90.9|82|81.4|
> > >     |Orig.|85.6|67.1|71.3|94|38.2|65.8|80.2|61.9|70.5|
> > >     |MASS|85.9|71|74.7|95.7|42|69.6|83.6|71.6|74.3|
> > >
> > >     As we expected and explained in our rebuttal to reviewer NxxW, performance declines after CPT because even MASS-filtered data likely has lower quality than the industry-standard data used in Qwen2.5. Nevertheless, MASS still outperforms the original data, demonstrating its effectiveness.
> > >
> > > [1] Programming Every Example: Lifting Pre-training Data Quality Like Experts at Scale, ArXiv 2024.
> > >
> > > [2] https://huggingface.co/datasets/gair-prox/open-web-math-pro
> > >
> > > [3] https://github.com/dorianbrown/rank_bm25

---

### Official Review · Reviewer_NxxW · 2025-03-19

**Overall Recommendation:** 3

**Summary:**

This paper introduces a method for math data selection in pre-training. It begins by extracting a skill graph from a high-quality reference dataset, then utilizes this graph to score a larger dataset and filter out high-quality samples.

**Claims And Evidence:**

Yes

**Essential References Not Discussed:**

No.

**Experimental Designs Or Analyses:**

Yes, the experiments on continued pre-training are well designed. However, in addition to the evaluation concerns mentioned earlier, it might be better to use stronger base models, such as DeepSeek-Code, which serves as the starting point for DeepSeek-Math's continued pre-training. Other 7B models stronger than Mistral is also better.

**Methods And Evaluation Criteria:**

Yes, they evaluated the models before and after continued pre-training on various math benchmarks. However, it would be beneficial to also include results on:

1. Corresponding results after instruction tuning for each model.
2. Performance on additional tasks, such as general instruction following and coding, to provide a more comprehensive assessment.

**Other Comments Or Suggestions:**

No.

**Other Strengths And Weaknesses:**

Strength:

1. The writing is clear and concise, without unnecessary information.
2. They conduct extensive ablation studies.
3. The method seems to be simple and practical.

Weakness:

1. It would be better to use stronger base model with larger-scale training and more comprehensive evaluation.

**Questions For Authors:**

1. How can this method be adapted for other data domains?

2. How would higher-quality math data impact the overall model capabilities?

**Relation To Broader Scientific Literature:**

This paper presents a practical and straightforward pipeline for pre-training data selection, particularly for math. It would be interesting to see this approach applied to stronger base models and larger-scale training.

**Theoretical Claims:**

There are no proofs.

---

> ### Author Rebuttal · Authors · 2025-04-01
>
> Dear Reviewer NxxW,
>
> Thank you for your thoughtful feedback and positive recognition. Below, we respond to each of your comments in detail.
>
> 1. **Weakness 1.1:** It would be better to use **stronger base models.**
>
>     **A:** Thank you for your suggestion. We chose Qwen2.5-7B as a stronger base model, and are currently pretraining it on both original OpenWebMath and MASS-selected OpenWebMath. The training is still in progress, and we will share the results in the discussion phase once the training is complete.
>
>     Additionally, we would like to explain why we did not opt for these SOTA models in the first place. The datasets we use are all open-sourced and have probably already been used in the pre-training stage of these strong models. For instance, Qwen2.5 was trained on 18 trillion tokens [1]. So, even if we selected high-quality data from OpenWebMath and continue pretraining Qwen2.5 on it, we would likely observe minimal to no performance improvements. Conversely, a relatively smaller and less capable model might be more suitable to test the data selection methods effectively.
>
> 3. **Weakness 1.2:** More comprehensive evaluation: It would be beneficial to include results on corresponding results after **instruction tuning**.
>
>     **A:** Thank you for your suggestion, but we would like to emphasize the three reasons why MASS prioritizes the data selection of pre-training over instruction-tuning and did not provide results after instruction-tuning:
>
>     - First, pre-training datasets are typically much larger than instruction-tuning datasets (e.g., OpenWebMath with 14.7 billion tokens vs. MetaMathQA with 103 million tokens). Due to their scale, pre-training datasets often contain a significant amount of noisy, repetitive, and irrelevant low-quality data, whereas fine-tuning datasets are generally more compact and high-quality, gaining little benefit from selection.
>     - Second, most of the knowledge LLMs acquire comes from the pre-training stage, while the instruction-tuning stage primarily focuses on aligning with human preferences and formatting. Thus, refining the vast pre-training datasets through the lens of skills is more effective and directly impacts model performance.
>     - Third, in our experiments, we used the Jiuzhang dataset, which follows a QA format similar to an instruction tuning dataset but at a much larger scale. This successfully demonstrates MASS's effectiveness even in an instruction-tuning-like setting to some extent.
>
> 2. **Question 1:** It would be benificial to include results on performance on **additional tasks, such as general instruction following and coding**. / How can this method be adapted for **other data domains?**
>
>     **A:** Thank you for your suggestion. While we cannot fully implement the method across other domains during the rebuttal phase, our future direction is to adapt MASS to other tasks and domains.
>
>     Since the core of MASS is a skill graph so it would be naturally suitable for domains where distinct and clear 'skills' exist. For example, in the coding area, we may extract skills such as *['CSV file handling','data visualization','SQL query construction',...]* (generated from *iamtarun/python_code_instructions_18k_alpaca dataset* by DeepSeek). In the biomedical area, skills may include *['alcohol-related disorders','neurotoxic substance effects','thiamine deficiency',...]* (generated from *FreedomIntelligence/medical-o1-reasoning-SFT* dataset by DeepSeek).
>
>     After skill extraction, we can construct the corresponding skill graph and apply MASS’s data selection approach. Some minor adjustments (e.g., prompts, graph construction) may be needed for domain-specific adaptations. However, for domains like creative writing or role playing, where skills are harder to define, the current MASS framework may not be suitable.
>
> 4. **Question 2:** How would higher-quality math data impact the **overall model capabilities**?
>
>     **A:** We believe higher-quality math data can impact the model capability in two ways:
>     - Improving math ability: High-quality mathematical data can directly enhance a model's mathematical reasoning skills, just as shown in our manuscript.
>     - Improving general reasoning ability: mathematical data is inherently logical and structured, so it can also help models better grasp logical relationships in complex tasks such as scientific document processing and code generation. For example, pre-training on datasets containing mathematical proofs and formulas can improve a model's ability to handle tasks requiring logical reasoning.
>
>
> We thank you again for your thoughtful advice, which has strengthened our work. We will include relevant details and analysis in the next version of our manuscript. Should you have any further questions or require additional information, we are happy to address them.
>
> Sincerely,
>
> MASS authors
>
> [1] https://qwenlm.github.io/blog/qwen2.5/

---

### Decision · Program_Chairs · 2025-05-01

**Decision:**

Accept (poster)

**Comment:**

The paper proposes MASS, a novel framework for mathematical data selection using a skill graph to enhance LLM pretraining specifically in mathematical reasoning. The authors effectively demonstrate the method's efficiency, showing substantial reductions (50%-70%) in required training tokens without sacrificing performance. Moreover, models pretrained using MASS-selected data notably outperform those trained on original datasets by up to 5.9%.

The reviewers unanimously recognize the manuscript's strengths, highlighting its clear writing, practical novelty, and solid empirical validation. Reviewers in particular note the rigorous experimental design, including comprehensive ablation studies and scalability assessments across different models (1B and 7B). Additionally, the proposed skill graph method offers interpretability and adaptability, potentially extending to other structured domains such as coding or biomedical fields.

Weaknesses pointed out by reviewers include the need for stronger base model benchmarks, additional comparative evaluations after instruction tuning, and theoretical underpinnings explaining the superior performance clearly. The authors have adequately addressed these concerns through rebuttal.

Overall, the paper offers a compelling combination of empirical results, methodological innovation, and practical relevance. Given the positive feedback from reviewers and the thoroughness of the authors’ response, the submission is recommended for acceptance at ICML.